# Photocatalytic Degradation, Anticancer, and Antibacterial Studies of *Lysinibacillus sphaericus* Biosynthesized Hybrid Metal/Semiconductor Nanocomposites

**DOI:** 10.3390/microorganisms11071810

**Published:** 2023-07-14

**Authors:** Kannan Badri Narayanan, Rakesh Bhaskar, Yong Joo Seok, Sung Soo Han

**Affiliations:** 1School of Chemical Engineering, Yeungnam University, 280 Daehak-Ro, Gyeongsan 38541, Gyeongbuk, Republic of Korea; indiaxenobiotics@gmail.com (R.B.); badri.explore@gmail.com (Y.J.S.); 2Research Institute of Cell Culture, Yeungnam University, 280 Daehak-Ro, Gyeongsan 38541, Gyeongbuk, Republic of Korea

**Keywords:** silver/zinc oxide nanocomposite, semiconductor, microbial biosynthesis, photodegradation, cytotoxicity, antibacterial

## Abstract

The biological synthesis of nanocomposites has become cost-effective and environmentally friendly and can achieve sustainability with high efficiency. Recently, the biological synthesis of semiconductor and metal-doped semiconductor nanocomposites with enhanced photocatalytic degradation efficiency, anticancer, and antibacterial properties has attracted considerable attention. To this end, for the first time, we biosynthesized zinc oxide (ZnO) and silver/ZnO nanocomposites (Ag/ZnO NCs) as semiconductor and metal-doped semiconductor nanocomposites, respectively, using the cell-free filtrate (CFF) of the bacterium *Lysinibacillus sphaericus*. The biosynthesized ZnO and Ag/ZnO NCs were characterized by various techniques, such as ultraviolet-visible spectroscopy, X-ray diffraction, Fourier transform infrared spectroscopy, field emission scanning electron microscopy, transmission electron microscopy, and photoluminescence spectroscopy. The photocatalytic degradation potential of these semiconductor NPs and metal-semiconductor NCs was evaluated against thiazine dye, methylene blue (MB) degradation, under simulated solar irradiation. Ag/ZnO showed 90.4 ± 0.46% photocatalytic degradation of MB, compared to 38.18 ± 0.15% by ZnO in 120 min. The cytotoxicity of ZnO and Ag/ZnO on human cervical HeLa cancer cells was determined using an MTT assay. Both nanomaterials exhibited cytotoxicity in a concentration- and time-dependent manner on HeLa cells. The antibacterial activity was also determined against Gram-negative (*Escherichia coli*) and Gram-positive (*Staphylococcus aureus*). Compared to ZnO, Ag/ZnO NCs showed higher antibacterial activity. Hence, the biosynthesis of semiconductor nanoparticles could be a promising strategy for developing hybrid metal/semiconductor nanomaterials for different biomedical and environmental applications.

## 1. Introduction

The industrial revolution in the current world has produced numerous associated environmental pollutants, which is a great concern for humankind. In particular, the textile and dyeing industries and other industries where several organic dyes are used, as well as the release of dye-contaminated industrial effluents as refractory pollutants, contaminate nearby waterbodies and threaten humans and aquatic life [1]. Textile dye effluent is one of the major pollution problems in the world. These dye-contaminated effluents pollute soil, surface water, and groundwater [2]. Until the early 19th century, natural dyes were used for coloring textiles and other decorations. In 1856, William H. Perkin was the first to synthesize the synthetic organic dye “mauveine” while attempting to synthesize the antimalarial drug quinine [3,4]. Currently, several dyes are produced through azo coupling and are produced in several million tons every year globally. These dye effluents can hinder the photosynthetic activity of aquatic plants, algae, and phytoplanktons and reduce oxygen production, accelerating the impacts of climate change. It is estimated that 50–80% of oxygen production comes from the photosynthetic activity of phytoplanktons [5]. These synthetic refractory dyes must be degraded to protect the ecosystem from their toxic effects.

Generally, various physical and chemical methods such as adsorption, precipitation, coagulation–flocculation, electrolysis, ultrasound, reduction, photocatalysis, electrochemical treatment, the Fenton process, and ion exchange are used for the removal and degradation of dyes from industrial effluents [6,7]. Developing advanced oxidation processes (AOPs) using sunlight in a wide wavelength range to activate photocatalysis by the synthesized catalyst is advantageous; here, sunlight is a sustainable and limitless energy source for both homogeneous and heterogeneous photocatalysis. On the other hand, there are also biological methods using microbes such as *Brevibacillus* sps., *Brachymonas* sps., *Bacillus* sps., *Pseudomonas* sps., *Acinetobacter* sps. *Fusarium* sps. *Trichoderma* sps., mixed cultures, and genetically engineered microorganisms for the bioremediation of dyes from industrial effluents [8,9,10]. Although these biological methods are cost-effective and eco-friendly for the degradation of dyes, they are time-consuming. In this scenario, nanotechnology-driven methodologies can offer efficient solutions to managing dye-contaminated industrial effluents. The efficiency of nanomaterials is greatly based on the type of metal or metal oxide nanoparticles, their smaller dimensions, and their large surface-to-volume ratio [11]. The synthesis of these nanoparticles (NPs) with unique physicochemical and optical properties can also fall under the physical, chemical, and biological categories. Again, the biological synthesis of nanoparticles is preferred due to its adherence to green principles over other synthetic routes [12]. Thus, microbial-mediated syntheses of nanoparticles are advantageous. 

Metal or metal oxide nanoparticles (NPs) composed of silver [13], gold [14], zinc [15], copper [16], palladium [17], platinum [18], or ruthenium [19] are commonly used photocatalysts for environmental remediation applications. Among the various photocatalysts, zinc oxide (ZnO) is an excellent semiconductor photocatalyst that is also chemically stable, cost-effective, and non-toxic. However, the band gap energy of 3.37 eV limits the most efficient photocatalytic degradation of dyes by ZnO under ultraviolet (UV) light and is less efficient under visible wavelengths [20]. There have been several attempts to enhance the photocatalytic performance of ZnO, including increasing the specific surface area and refining the grain size [20,21,22]. Currently, element doping and recombination with metals in semiconductors producing nanocomposites (NCs) are commonly used to improve the photoresponse range of ZnO NPs as a photocatalyst. Kareem et al. [23] demonstrated the synthesis of silver-doped ZnO nanoparticles and their enhanced photocatalytic ability towards methylene blue (MB) degradation compared to undoped ZnO. The incorporation of Ag-doped ZnO nanorods through graphite hybridization (ZnO-Ag-Gp) exhibited degradation of the pharmaceutical pollutant ciprofloxacin [24]. In addition, there is also a need to develop antibacterial materials to deal with the emergence of antibiotic-resistant bacterial strains. ZnO NPs were proven to exhibit strong antibacterial properties, and the Food and Drug Administration (FDA) has approved using ZnO material for biomedical applications [25]. Incorporating Ag NPs with ZnO was also used to formulate various pharmaceutical products for skin infections and composite materials for biomedical applications [26]. Ag/ZnO nanocomposites also exhibit high cytotoxicity against various cancer cells. Cancer cells are different from normal cells in their metabolism, and Zn^2+^ can induce reactive oxygen species (ROS) production in cancer cells, causing death [27]. ZnO NPs are also reported to cause the death of cancer cells by altering the post-translational modification of histones by methylation [28]. Intriguingly, silver is well known to form free radicals, which induce apoptosis in cancerous cells [25]. In our study, we demonstrated, for the first time, the biosynthesis and characterization of zinc oxide (ZnO) as semiconductor particles and silver/ZnO (Ag/ZnO) as hybrid metal/semiconductor NCs using the cell-free filtrate (CFF) of *L. sphaericus*. These synthesized materials were evaluated for their photodegradation efficiency of a thiazine dye, in vitro cytotoxicity efficiency against HeLa cancer cells, and antibacterial properties against Gram-positive and Gram-negative bacteria.

## 2. Materials

Zinc nitrate hexahydrate (Zn(NO_3_)_2_·6H_2_O) (99% extra pure), dipotassium phosphate (K_2_HPO_4_), and methylene blue (MB) were obtained from Duksan Pure Chemicals Co., Ltd. (Ansan-si, South Korea). *Lysinibacillus sphaericus* (ATCC 14577) was purchased from the American Type Culture Collection (Manassas, VA, USA). *Escherichia coli* (KCTC 2571) and *Staphylococcus aureus* (KCTC 3881) were obtained from the Korean Collection for Type Cultures (Jeongeup, Republic of Korea). Malt extract, tryptone peptone, yeast extract, soytone, and agar were purchased from Difco^TM^ (Detroit, MI, USA). MTT (3-(4,5-dimethylthiazol-2-yl)-2,5-diphenyltetrazolium bromide) and silver nitrate (99% extra pure) were purchased from Sigma-Aldrich (St. Louis, MI, USA). The human cervical cancer cell line (HeLa) (CCL-2) was purchased from the American Type Culture Collection (ATCC) (Manassas, VA, USA). Eagle’s minimum essential medium (EMEM), fetal bovine serum (FBS), penicillin/streptomycin, 0.25% trypsin-EDTA, and 1% penicillin/streptomycin were purchased from Gibco^TM^ (Thermo Fisher Scientific, Waltham, MA, USA). Deionized water collected from the Milli-Q^®^ Direct water purification system (Merck Millipore, Darmstadt, Germany) was used for experiments.

### 2.1. Culturing of the Lysinibacillus sphaericus Bacterium

Bacterium *Lysinibacillus sphaericus* was cultured on tryptic soy agar (TSA), and the purified bacterium was grown on TS broth (containing 17 g/L tryptone peptone, 3 g/L soytone, 5 g/L NaCl, 2.5 g/L glucose, and 2.5 g/L K_2_HPO_4_) at 30 °C for 16 h on a rotary shaker [29]. A total of 1% (*v*/*v*) of this culture was used as an inoculum to culture 100 mL of TS medium in a 250 mL Erlenmeyer flask for 16 h. Later, the culture was centrifuged to pellet bacterial biomass at 4000 rpm for 20 min at 4 °C, and the supernatant was filtered through a nylon filter membrane with a 0.45 µm pore size and stored in a sterile amber bottle at 4 °C for further experiments. 

### 2.2. Biosynthesis of Semiconductor and Metal/Semiconductor NCs

Zinc oxide (ZnO) particles were prepared using cell-free filtrate (CFF) of *L. sphaericus*. Briefly, 250 mL of zinc nitrate (0.1 M) was used as a zinc precursor solution and taken in a 1000 mL Erlenmeyer flask, and 250 mL of bacterial filtrate was added dropwise while stirring at 40 °C for 1 h. This mixture was further stirred at 40 °C for 72 h. In another batch, to prepare silver/ZnO (Ag/ZnO) NCs, zinc nitrate (0.1 M) was mixed with silver nitrate (0.1 mM) and stirred at 40 °C for 72 h. Bacterial filtrate without a zinc precursor solution was used as a negative control. After 72 h of incubation, the synthesized ZnO as semiconductor particles and Ag/ZnO as metal/semiconductor NCs were centrifuged at 10,000 rpm for 20 min, and the pellet was washed with deionized water and ethanol and dried overnight at 60 °C before being calcined at 450 °C for 6 h (Figure 1). These biosynthesized ZnO and Ag/ZnO NCs were further characterized for their physicochemical properties.

### 2.3. Characterization of ZnO and Ag/ZnO NCs

The biosynthesized ZnO and Ag/ZnO were characterized by ultraviolet-visible spectroscopy using a UV-vis-NIR spectrophotometer (Varian CARY 5000, Agilent Technologies, Santa Clara, CA, USA) between wavelengths 200 and 1100 nm. The functional groups of the CFF and their involvement in the synthesis of ZnO were analyzed by Fourier-transform infrared (FT-IR) spectroscopy (Model: Spectrum 100; Perkin Elmer, Waltham, MA, USA) in the 400–4000 cm^−1^ range. The hydrodynamic size and the zeta potential were analyzed using a Zeta sizer (Model: Nano ZS90; Malvern Instruments, Malvern, UK). The crystallinity of ZnO was assessed by powder X-ray diffraction analysis (XRD) using PANalytical X’PertPRO MPD (Eindhoven, The Netherlands) scanned at 2θ values of 10° to 90° operated at 30 mA and 40 kV with a radiation source of Cu Kα (λ = 1.54 Å). The crystallite size of ZnO was calculated by the Debye–Scherrer equation as follows: D = 0.9λ/βcosθ, where D is the crystallite size, 0.9 is the Scherrer’s constant, λ is the X-ray wavelength, β is full width at half maximum (FWHM), and θ is the Bragg diffraction angle (degrees). The morphology and the elemental composition of the ZnO and Ag/ZnO were analyzed using a Field emission scanning electron microscope (FE-SEM) coupled to energy dispersive spectroscopy (EDX) (Model: S4800; Hitachi, Japan). To observe the high-resolution morphology and elemental composition of NPs, field emission-transmission electron microscopy (FE-TEM) with energy-dispersive X-ray spectroscopy (EDX) was performed using Tecnai G^2^ F20 Twin TMP (Philips/FEI Company, Hillsboro, OR, USA) at an accelerating voltage of 200 kV. Photoluminescence (PL) spectroscopy was carried out using the Raman System (HORIBA Scientific, Kyoto, Japan) with a 325 nm He-Cd laser at 50 mW, and LabSpec 6 software (https://www.horiba.com/int/scientific/products/detail/action/show/Product/labspec-6-spectroscopy-suite-software-1843/ accessed on 22 June 2023) was used for analysis.

### 2.4. Evaluation of the Antibacterial Activity of ZnO and Ag/ZnO NCs

The antibacterial activity of ZnO and Ag/ZnO against Gram-positive (*Staphylococcus aureus*) and Gram-negative (*Escherichia coli*) bacteria was assessed using the agar well diffusion method and colony count method [30]. For the agar well diffusion method, overnight-cultured bacterial cultures of *S. aureus* and *E. coli* cultured on TS broth were spread-plated onto Muller Hinton (MH) agar medium, and agar wells were made using a sterile cork borer (8 mm). Different concentrations of biosynthesized ZnO and Ag/ZnO (200 and 400 µg; 40 µL) were added and incubated at 37 °C for 16 h. Ampicillin was used as a positive control, the zone of inhibition (ZOI) was measured, and the antibacterial activity was recorded. For the colony counting method, bacteria (3 × 10^8^ CFU/mL) were exposed to ZnO and Ag/ZnO (200 and 400 µg/mL) and incubated at 37 °C for 3 h. The live bacterial colonies were counted by spread plating on TSA, and colony-forming units were calculated.

### 2.5. In Vitro Cytotoxicity Assay

#### 2.5.1. Cell Culture

The HeLa cancer cell line was cultured in a 75 cm^2^ flask using EMEM supplemented with 10% FBS (*v*/*v*) and 1% penicillin/streptomycin (*w*/*v*) at 95% humidified atmosphere and 5% CO_2_ at 37 °C. At 80% confluence, cells were harvested by trypsinization using 0.25% trypsin-EDTA and counted using trypan blue staining in a hemocytometer.

#### 2.5.2. MTT Assay

The in vitro cytotoxicity of ZnO and Ag/ZnO against the HeLa cancer cell line was determined using the MTT assay. The HeLa cells (10,000 cells/well) were seeded onto 96 well plates and cultured for 3 days on an EMEM medium. After culturing, the old media was aspirated and replenished with new media containing different concentrations of ZnO and Ag/ZnO (1.56–100 µg/mL) and exposed for 24 and 48 h under specified conditions. After the treatment, MTT solution (0.5 mg/mL) was added to each well and incubated in the dark for 4 h. The untreated wells were used as controls. Then, the supernatant was removed, the formazan was dissolved with DMSO, and the absorbance was read at 570 nm with a reference wavelength of 690 nm using an Epoch^TM^ microplate spectrophotometer (BioTek Instruments, Winooski, VT, USA). To determine the in vitro cytotoxicity, we calculated the percent cytotoxicity as % Cytotoxicity = 100 × (OD value of control − OD value of sample)/OD value of the control.

### 2.6. Photocatalytic Degradation of Methylene Blue Using ZnO and Ag/ZnO NCs

The photocatalytic degradation of methylene blue (MB) by biosynthesized ZnO and Ag/ZnO as photocatalysts was assessed under simulated solar light irradiation [1]. Briefly, 100 mL of 0.001% (*w*/*v*) MB dissolved in deionized water was taken in a 250 mL conical beaker, and 0.1% (*w*/*v*) ZnO or Ag/ZnO was added and incubated in the dark at room temperature for 30 min to achieve adsorption–desorption equilibrium. Later, the mixture in a conical beaker was exposed to simulated solar irradiation using an Ultra-Vitalux lamp (300 W) at a 15 cm distance. For 120 min, samples were taken and centrifuged at 12,000 rpm for 10 min, and the absorbance of the supernatant was analyzed using a UV-vis spectrophotometer. The concentration of MB was quantified using the maximum absorbance (λ_max_) of the solution at 665 nm. The degradation percentage of MB was calculated using the dye concentration after a particular time over the initial dye concentration as follows: Degradation (%) = ((C_0_ − C_t_)/C_0_) × 100, where C_0_ and C_t_ are the concentrations of dye after the dark period of adsorption–desorption equilibrium and at specific time intervals, respectively.

## 3. Results and Discussion

### 3.1. Mechanism of Biological Synthesis of ZnO and Ag/ZnO NCs

The biological synthesis of metal oxide NPs and metal/metal oxide NCs as semiconductor NPs and metal/semiconductor NCs can be achieved through plants, microorganisms, secondary metabolites, and biomacromolecules. The biological synthesis of these NPs in a greener method without toxic or hazardous chemicals at ambient conditions is sustainable and eco-friendly. The bottom-up approach of arranging atoms into nuclei, followed by the formation of nanoparticles, is widely used. Among different greener routes, microbe-mediated extracellular synthesis involving the oxidation or reduction of metallic ions by bacterial or fungal enzymes and their biomolecules is more advantageous. Oxidoreductase enzymes such as NADH-dependent nitrate reductases are mainly involved in the biocatalytic synthesis of metal or metal oxide NPs; the reaction between the bioreducing enzymes/biomolecules and Zn^2+^ ions from the metal ion precursor forms ZnO [12]. The silver nanoparticles (Ag NPs) are formed by the reduction of Ag^+^ ions by the proteins/sugars and other biomolecules of CFF on the surface of ZnO, forming Ag/ZnO NCs [31]. Various biomolecules, such as sugars and proteins, can also act as stabilization agents in forming Ag/ZnO NCs.

The formation of biosynthesized ZnO and Ag/ZnO was initially confirmed by visual assessment. The color of the reaction mixture was changed from a clear pale-yellow color to a turbid pale yellow-whitish color for ZnO and a brownish color for Ag/ZnO. These characteristic colors indicate the formation of ZnO precipitate and Ag NPs on ZnO. The enzymes, biomacromolecules, and bioactive metabolites on the CFF of *L. sphaericus* are presumed to be involved in forming and stabilizing ZnO and Ag/ZnO NCs (Figure 1). Jain et al. [32] demonstrated the synthesis of white precipitate ZnO NPs using zinc-tolerant bacteria *Serratia nematodiphila* by reduction process [32]. The rhamnolipids (RLs) of *Pseudomonas aeruginosa* and the phycocyanin pigment of cyanobacteria are also involved in synthesizing ZnO NPs [33,34]. The metabolites of *Bidens pilosa* and *Verbascum speciosum* plant extracts are also involved in the green synthesis of Ag/ZnO nanocomposites with antimicrobial and anticancer activities [25,35].

### 3.2. Characterization of Biosynthesized ZnO and Ag/ZnO NCs

Figure 2a shows the UV-vis spectra of ZnO and Ag/ZnO NCs. When treating the metal precursors with CFF, zinc ions form ZnO, and Ag^+^ ions form Ag NPs by reacting with bacterial enzymes and the bioactive molecules. The formation of ZnO and Ag/ZnO was investigated by UV-vis diffuse reflectance spectroscopy (UV-vis DRS). ZnO shows a maximum absorption band in the UV region at 366 nm, which agrees with the previous reports [1]. The formation of Ag NPs on ZnO in the production of Ag/ZnO was confirmed by the broad absorbance in the visible region for Ag NPs in the range of 450 to 550 nm. The formation of Ag NPs on ZnO causes a broad absorption band in the visible region due to the surface plasmon resonance (SPR) of polydispersed anisotropic Ag NPs [36]. The excitation of the Ag/ZnO causes the generation of more electron–hole pairs, resulting in the broadening of the absorption peak [37]. The intensity and position of the SPR band of Ag NPs are correlated with their size, shape, composition, and local environment. Similar results were reported in the biogenic synthesis of Ag/ZnO using the extracts of *Bidens pilosa* and *Crataegus monogyna* [35,38].

The band gap energy (E_g_) is the energy required to excite electrons from the valence band to the conduction band. The direct band gap energies of biosynthesized ZnO and Ag/ZnO are calculated using the Tauc equation: αhν = A (hν − E_g_)^n^; where α is the optical absorption coefficient, *h* is the Planck constant, ν is the photon’s energy, A is the constant, and Eg is the band gap energy. The E_g_ is determined using the optical absorption coefficient from the experimental absorbance. The extrapolation of the linear region of the curve to the *X*-axis gives the E_g_ values [39]. Figure 2b shows the E_g_ values of ZnO and Ag/ZnO. The optical band gap energies of biosynthesized ZnO and Ag/ZnO were 2.96 and 2.75 eV, respectively. ZnO can absorb light less than 400 nm, mainly in the UV region, hindering its visible light photocatalytic applications. With the binding of Ag NPs on the surface of ZnO, there was a decrease in the band gap energy of Ag/ZnO, which utilized the visible light absorption of more photons. The heterogeneous structure of Ag NPs on ZnO allows visible-light-induced activation due to the localized SPR of Ag NPs and enables the absorption of the whole solar spectrum. Thus, the band gap modification of ZnO through silver doping extends the absorption of light at the visible wavelength. The high intensity and broad absorption peak of Ag/ZnO through the entire solar spectrum can also facilitate enhanced light absorption capacity both in the UV and visible regions for photocatalytic activities [40]. The band gap energy of Ag/ZnO is lower than that of ZnO. The decrease in optical band gap energy can be influenced by grain size, carrier concentration, structural parameters, lattice strain, and defects or impurities [41,42].

The crystallite size, crystallinity, and purity of nanocomposites were investigated by XRD analysis. Figure 3a shows the XRD pattern of biosynthesized ZnO and Ag/ZnO. The XRD peaks of biosynthesized ZnO appeared at 31.7°, 34.39°, 36.28°, 47.60°, 56.57°, 62.81°, 66.44°, 68.08°, 69.37°, and 72.7° corresponding to (100), (002), (101), (102), (110), (103), (200), (112), (201), and (004) lattice of the hexagonal phase of wurtzite ZnO (JCPDS No. 036-1451) without any impurities after calcination [43,44]. The presence of Ag NPs in Ag/ZnO was confirmed by the diffraction peaks at 38.10°, 44.29°, 64.42°, 77.38°, and 81.52° corresponding (111), (200), (220), (311), and (222) to the face-centered cubic (fcc) silver phase (JCPDS number 2-109), which confirms the presence of silver along with the diffraction peaks of ZnO on the Ag/ZnO [1]. The silver ions (0.122 nm) are larger than the zinc ions (Zn^2+^); silver ions cannot substitute in the ZnO matrix; thus, they can only be formed over the surface of ZnO [45]. The average crystallite sizes of ZnO and Ag NPs in Ag/ZnO calculated using the Debye–Scherrer equation were 25.6 ± 9.9 nm and 24.9 ± 1.9 nm, respectively. Similarly, the crystallinity of ZnO and Ag/ZnO was 93.53% and 93.48%, respectively. The decrease in the intensity of diffraction peaks of ZnO in Ag/ZnO indicated a slight reduction in the crystalline structure of Ag/ZnO.

The presence of functional groups from the CFF of *L. sphaericus* used in the biosynthesis and stabilization of ZnO and Ag/ZnO was investigated using FTIR analysis in the range of 400–4000 cm^−1^ (Figure 3b). The infrared absorption spectra of ZnO and Ag/ZnO showed similar spectra, and the stretching vibration of the O–H bond of the H_2_O molecule in the Zn–O lattice around 3400 cm^−1^ almost disappeared after calcination [46]. The absorption bands around 2330 and 2345 cm^−1^ could have resulted from CO_2_ adsorption on the surface of ZnO and Ag/ZnO, respectively. The peak around 1620 cm^−1^ in ZnO was ascribed to H–O–H bending vibrations due to the water molecule [47]. The intense band at 960 cm^−1^ was correlated to the C–O stretching [48]. The broad IR band around 400–600 cm^−1^ was attributed to the metal–oxygen bond’s stretching vibration, confirming the formation of ZnO bonds [49]. 

Dynamic light scattering (DLS), also known as photon correlation spectroscopy, measures the Brownian motion of particles in solution and relates it to the size of the particles. DLS is a rapid technique to find the average size and distribution of NPs. Generally, DLS measures the hydrodynamic radius, which is influenced by the structure, shape, and surface properties of the NPs. Ag/ZnO shows an increase in the size of the NPs compared to ZnO, which could be due to Ag NPs forming on the surface of ZnO and by aggregation. Zeta (ζ) potential measurements are used to evaluate the surface charges of NPs. The NP’s stability is directly correlated to the magnitude of the zeta potential charge. ZnO and Ag/ZnO dispersed in an aqueous solution were used for the zeta potential measurements. The zeta potential values of colloidal particles are directly correlated with their stability. The higher zeta potential value indicates better physical colloidal stability. The colloidal particles with ζ values between ±10 and ±30 are considered incipiently stable, and those with ±30 and ±40 are considered moderately stable. The zeta potential values of ZnO and Ag/ZnO were −30.1 ± 8.39 and −29 ± 5.74 mV, respectively, indicating that the negatively charged biomacromolecules of the CFF of *L. sphaericus* are involved in stabilization (Appendix A).

FE-SEM was used to observe the surface morphology of ZnO and Ag/ZnO. ZnO synthesized by the direct precipitation method using the CFF of *L. sphaericus* as an additive promoted the formation of a puffy-like morphology (Figure 4a–d). The addition of sustainable and eco-friendly materials as additives directs the formation of ZnO with a unique morphology [50]. The calcination process at different temperatures also influences the formation of different morphologies [51]. The puffy-like morphology of ZnO was anisotropic and polydispersed and aggregated nanoparticles. The agglomeration of ZnO was presumed to have been caused by the higher surface area and affinity among particles. In the case of Ag/ZnO, the addition of silver nitrate to the CFF-treated zinc nitrate mixture resulted in the formation of irregularly shaped polydispersed ZnO with nano-dimensioned Ag NPs on their surfaces.

The FE-SEM-EDX analysis was performed to investigate the elemental composition of ZnO and Ag/ZnO. The EDX spectra and the elemental mapping with the quantified elemental composition of ZnO and Ag/ZnO are shown in Appendix A and Figure 4e–i, respectively. The EDX spectrum of ZnO showed an oxygen (O-Kα) peak at low energy (0.560 keV) and Zn-Lα, Zn-Kα, and Zn-Kβ peaks of the zinc element at ~1.01, 8.6, and 9.5 keV, respectively. At the same time, Ag/ZnO spectra showed the presence of silver element peaks, viz. Ag-Kα, Ag-Lα, Ag-Lβ, and Ag-Lβ2 peaks at ~2.6, 3.1, 3.26, and 3.35 keV, respectively, along with the Zn and O peaks, which infers that silver was successfully incorporated as Ag NPs in forming Ag/ZnO NCs.

Figure 5a–c show the FE-TEM images of biosynthesized Ag/ZnO at different magnifications. FE-TEM images verified the formation of puffy-like aggregated ZnO nanoparticles and the attachment of Ag NPs on the surface of ZnO. It was found that nanosized spherical and quasi-spherical Ag NPs (dark spots) with a size of 19.0 ± 11.9 nm were attached to ZnO.

A high-angle annular dark field (HAADF) STEM image of the biosynthesized Ag/ZnO shows that Ag NPs are formed on the surface of ZnO. The elemental composition analysis by EDX line scan confirmed the presence of silver on the presumed Ag NPs particles on ZnO (Figure 6a–d). 

Photoluminescence (PL) is an essential technique for analyzing biosynthesized ZnO and Ag/ZnO due to their visible PL nature. ZnO interacts with light and undergoes photogeneration of electron–hole (e^−^/h^+^) pairs that recombine radiatively, giving rise to PL spectra. The emission spectrum of ZnO comprises near-band edge (NBE) emission and deep-level emission (DLE). The defects or foreign impurities in the ZnO can change the PL outcome. Figure 7a shows the PL spectra of ZnO and Ag/ZnO at room temperature. A typical peak in the visible region of ~480 nm comes from oxygen vacancies and other defects in ZnO nanostructures [52]. Both show the predominant emission only in the DLE mode around 400–700 nm and almost no emission in the NBE region below 400 nm. The low NBE peak confirms that ZnO is granularly textured and exhibits low optical quality. The presence of DLE indicates the presence of deep-level radiative defects [53]. The addition of silver to ZnO results in the formation of Ag/ZnO NCs with increased e^−^/h^+^ lifetime and a subsequent reduction in their recombination rate [54,55]. Also, the intensity of the broad visible defect emission decreases with the addition of Ag with ZnO, which also indicates the separation of photoinduced e^−^/h^+^ pairs and prolonged recombination of photoinduced pairs due to the metal–semiconductor diode effect between Ag NPs and ZnO in Ag/ZnO [56]. Ag/ZnO NCs produced by goji berry extract show decreased PL intensity in the UV emission compared to ZnO, indicating a decrease in the e^−^/h^+^ recombination rate and efficient charge separation [1]. 

### 3.3. Applications of Biosynthesized ZnO and Ag/ZnO NCs

#### 3.3.1. Antibacterial Assay

The antibacterial activity of biosynthesized ZnO and Ag/ZnO NCs is shown in Figure 8a,b. The agar well-diffusion method shows that both Gram-negative (*E. coli*) and Gram-positive (*S. aureus*) bacteria were inhibited largely by Ag/ZnO NCs, and ZnO did not show any profound antibacterial activity. Moreover, *E. coli* showed slight resistance to Ag/ZnO compared to *S. aureus*. The antibacterial activity of NPs could also be influenced by their shape, size, surface charge, and surface functionalization [57,58]. The ZOIs for Ag/ZnO were 9.95 ± 0.588 (200 µg), 10.35 ± 0.31 (400 µg), and 12.12 ± 0.65 (200 µg), 12.48 ± 0.509 (400 µg) towards *E. coli* and *S. aureus*, respectively, whereas the ZOI of control was 12.18 ± 1.47 (500 µg) and 31.42 ± 1.05 mm (300 µg) for *E. coli* and *S. aureus*, respectively. The colony-forming units (CFU) assay method was used to evaluate the effect of biosynthesized ZnO and Ag/ZnO NCs on *E. coli* and *S. aureus* in solution. After 3 h of exposure to different concentrations of NPs (250–62.5 µg/mL), it was found that Ag/ZnO showed higher antibacterial activity compared to ZnO (Figure 9c,d). All concentrations of NPs showed significant antibacterial activity against both *E. coli* and *S. aureus*. Compared to *E. coli*, *S. aureus* showed highly significant susceptibility to the antibacterial activity of Ag/ZnO NCs at all concentrations. Nevertheless, at a concentration of 250 µg/mL, Ag/ZnO NCs showed more complete bactericidal activity in both *E. coli* and *S. aureus* than ZnO. The antibacterial activity of Ag/ZnO was due to the synergistic effect of Ag and ZnO as well as the production of high levels of ROS generation [59,60]. The oxidative stress caused by ROS generation also plays a crucial role in inhibiting bacterial growth. The interaction of metal nanoparticles with the bacterial membrane results in the loss of membrane integrity and the oxidation of membrane lipids, causing adverse effects on the bacteria. Moreover, Ag NPs or Ag/ZnO also trigger the production of hydroxyl radicals and singlet oxygen, causing aberrations in the cell membrane components and leading to bacterial death [7].

#### 3.3.2. Photocatalytic Degradation

The photocatalytic efficiency of biosynthesized ZnO and Ag/ZnO in methylene blue (MB) degradation, the thiazine dye, has been investigated under simulated solar light. Figure 9a,b show the UV-vis absorption spectra of the photodegradation of MB with biosynthesized ZnO and Ag/ZnO with time. In the presence of a catalyst, the cationic thiazine dye, MB, was reduced into the less toxic colorless leuco-methylene blue (LMB) [61]. The decolorization of the dye solution is a visible confirmation of the degradation of the dye. UV-vis spectra of the reaction mixture containing dye and biosynthesized ZnO or Ag/ZnO irradiated with simulated solar light were recorded at different intervals. The degradation of MB dye was monitored with the decrease in the intensity of the λ_max_ (665 nm) of the MB. The efficiency of photocatalysis is dependent on the properties of the photocatalyst material. The photocatalysis occurs on the surface of the photocatalyst; therefore, increasing the surface-to-volume ratio, manipulating the band structure of the photocatalyst, and limiting the recombination of photogenerated pairs can improve the photocatalytic performance of the photocatalyst nanomaterial [54,62,63]. Figure 9c shows the photocatalytic degradation graph of MB with time by ZnO and Ag/ZnO, i.e., C_t_/C_0_ versus time, where C_0_ and C_t_ are the concentrations of MB after the dark period of adsorption–desorption equilibrium and time intervals “t,” respectively. The degradation of MB without catalyst was also evaluated; it shows that the MB concentration was slowly reduced by photoreaction, i.e., degradation of 25.8 ± 1.91% after 180 min (Appendix A). However, in contrast, the degradation of MB was dramatically increased in the presence of Ag/ZnO as a photocatalyst. The presence of plasmonic NPs on the ZnO increased the photocatalytic activity of Ag/ZnO NCs.

The reaction mixture containing photocatalyst (0.1% *w*/*v*) and MB (1 mg/100 mL) was incubated in the dark to reach adsorption–desorption equilibrium for 30 min, which shows that MB concentrations of 20.29 ± 0.91% and 2.32 ± 1.32% were adsorbed on the photocatalysts ZnO and Ag/ZnO, respectively. The incorporation of silver into the ZnO decreased the adsorption of MB to the surface of Ag/ZnO more than ZnO. However, despite the decrease in the adsorption of MB by Ag/ZnO in the dark, there was a significant increase in its photocatalytic degradation capacity [64]. Ag/ZnO showed 90.4 ± 0.46% photocatalytic degradation, which is more robust than ZnO with a 38.18 ± 0.15% photodegradation potential for 120 min (Figure 9d). The photocatalytic degradation of MB by Ag/ZnO was significantly higher than ZnO, and the enhanced degradation was due to the presence of Ag as a dopant in the ZnO lattice. The addition of silver enhances the photodegradation potential of Ag/ZnO. The addition of Ag NPs as a dopant on the ZnO acts as an electron trapper. It decreases the recombination capacity of photogenerated electron–hole pairs, thereby increasing carrier lifespan and enhancing photocatalytic efficiency [65]. The reusability of the catalyst is an important parameter required for photocatalytic reaction. Due to the challenges in separating catalysts from the medium following the degradation of contaminants, the use of powdered catalysts has been restricted. However, modern catalysts are easily separated from the solution by filtration or centrifugation, and dyes do not adhere permanently to the photocatalyst. In our study, after each reaction, the Ag/ZnO was collected by centrifugation, washed three times with deionized water, and regenerated by drying it at 100 °C for 1 h after washing it in ethanol. The photocatalytic activity of Ag/ZnO in the degradation of MB remains significantly higher for up to five cycles, and there was only an indiscernible decrease of 7.3% in the degradation ability from the initial degradation capability (Appendix A).

Figure 7b depicts the most probable photocatalytic mechanism for the degradation of MB by biosynthesized Ag/ZnO under simulated solar irradiation. When Ag NPs are decorated on the surface of ZnO in Ag/ZnO, the conduction band (CB) becomes slightly positive. The activation of Ag/ZnO by incident photon energy causes the excitation of electrons in ZnO from the VB to the CB. The Fermi level (E_FM_) of Ag is more positive than that of ZnO (E_FS_), which leads to the bending of the energy band due to the injection of electrons from the CB of ZnO to Ag [66]. This creates a new Fermi level equilibrium (E_F_) between the Fermi levels of ZnO and Ag. The migration of electrons from ZnO to Ag creates a Schottky barrier at the interface between them, preventing the recombination of electron–hole (e^−^/h^+^) pairs. The visible light of the incident light also causes the localized SPR effect on Ag NPs, which excites Ag electrons to a higher excitation level and causes migration to the CB of ZnO due to the energy level difference. This also significantly hinders the recombination of e^−^/h^+^ pairs and causes the accumulation of electrons on the CB of ZnO and holes on the VB of ZnO and Ag NPs. The generation of electron–hole pairs (e^−^/h^+^) reacts with molecular oxygen (O_2_) and water molecules, resulting in the production of reactive oxygen species (ROS), such as superoxide anion radicals (^•^O_2_^−^) and hydroxyl radicals (^•^OH), which are involved in the degradation of cationic thiazine dye, MB, into CO_2_ and minerals [67]. Ha et al. [65] demonstrated that the photogenerated electrons (e^−^) by Ag/ZnO prompted the photocatalytic efficiency of the nanomaterial due to the Schottky barrier formation of the composite and the localized SPR effect of the anisotropic Ag NPs. The photogenerated holes (h^+^) also react with water molecules to yield hydroxyl radicals, which also involves the photocatalytic degradation of MB.

#### 3.3.3. In Vitro Cytotoxicity Assay

Due to their excellent transport and increased permeability and retention (EPR) properties, nanoparticles with a diameter of 10–100 nm are regarded as effective anticancer agents. Smaller particles (<10 nm in size) are thought to be rapidly released from blood vessels and damage healthy cells and tissue before being metabolized by the kidneys [68]. Exposure to all concentrations of biosynthesized ZnO and Ag/ZnO (100–1.56 µg/mL) significantly induced cytotoxicity in a concentration- and time-dependent manner on HeLa cancer cells. The highest cytotoxicity for ZnO was 42.26 ± 2.23% on HeLa, whereas Ag/ZnO exhibited 50.5 ± 0.15% at a concentration of 100 µg/mL after 48 h of exposure (Figure 10I). Compared to Ag/ZnO, ZnO was found to be less cytotoxic against HeLa cancer cells. The elevated cytotoxicity of nanoparticles against HeLa cells with time and concentration occurs due to the highest uptake of nanoparticles by the cancer cells and their increased ROS generation, which inhibits the transcriptional process [69]. Silver nanoparticles of Ag/ZnO NCs were reported to cause cytotoxicity to cancer cells through lipid peroxidation, protein thiol oxidation, and necrosis. In contrast, silver ions can cause H_2_O_2_ elevation and drive oxidative stress and apoptotic cell death [70]. Biogenic ZnO NPs synthesized using *Pandanus odorifer* leaf extract exhibited cytotoxicity against various cancer cells (MCF-7, HepG2, and A-549) at a concentration of 50–100 µg/mL [71]. Chandrasekaran et al. [72] prepared ZnO NPs using *Vinca rosea* leaf extract, which showed potent cytotoxic activity against MCF-7 cells. Similarly, Tanino et al. [73] reported that the anticancer activity of ZnO NPs was exhibited by inducing highly reactive ROS, DNA leakage from nuclei by membrane rupture, apoptosis, and necrosis. The morphological properties of ZnO NPs are also extensively involved in their anticancer activity. It was found that spherical ZnO NPs at 100 µg/mL showed better anticancer activity than hexagonal and rod-like NPs [74]. In our study, Ag/ZnO NCs exhibited a strong anticancer activity compared to ZnO. Rafique et al. [26] also demonstrated that ZnO/Ag NPs produced by *Moringa oleifera* exhibited strong anticancer activity in HeLa cells. Moreover, cyclodextrin-Ag NPs decorated titanium dioxide (TiO_2_-Ag NPs @CD) also induced superior cytotoxicity at a concentration of 64 µg/mL on HeLa cells. The disintegration of cell membranes and the production of ROS, causing oxidative stress, are involved in the anticancer properties of these NCs [75].

Figure 10II shows the live (green) and dead (red) cells of HeLa cancer cells after exposure to biosynthesized ZnO and Ag/ZnO for 48 h. The combined use of calcein and propidium iodide in a dual fluorescence assay allows the evaluation of cellular viability and cell death of HeLa cancer cells. The microscopic fluorescent images of HeLa cells exposed to nanocomposites for 48 h concur with the results of the MTT assay. The nonfluorescent calcein AM dye was converted into green, fluorescent calcein dye by the intracellular esterase activity of the live cell. In contrast, the propidium iodide red fluorescence increased by intercalating into the nucleic acid of the dead cells [76].

## 4. Conclusions

In this paper, we successfully biosynthesized ZnO by the direct precipitation method, and Ag NPs on ZnO were produced by the reduction process involving biomacromolecules and enzymes of the CFF of *L. sphaericus* to prepare Ag/ZnO NCs in an environmentally benign and greener route. These biosynthesized ZnO were irregularly shaped, puffy-like aggregated hexagonal wurtzite crystal structure nanoparticles with an average crystallite size of 25.6 ± 9.9 nm. The spherical and quasi-spherical Ag NPs, with an average size of 19.0 ± 11.9 nm, were embedded on the surface of ZnO. These semiconductor NPs and metal/semiconductor NCs showed excellent antibacterial, anticancer, and photocatalytic degradation capabilities. The photocatalytic degradation efficiency of Ag/ZnO was higher than ZnO, exhibiting improved separation and lifetime of photogenerated electron/hole pairs. The addition of Ag to ZnO contributed to the increased decomposition of methylene blue, up to 90.4% in 120 min. These biosynthesized NPs exhibited significant antibacterial activity against *E. coli* and *S. aureus* and anticancer activity against HeLa cervical cancer cells. However, the average hydrodynamic particle size of Ag/ZnO was >1000 nm which can trigger macrophage immune response and limits its applications in systemic cancer therapy [77]. Thus, these biosynthesized materials could be used in localized cancer treatments. Overall, the use of microbial cell-free filtrate for the biological synthesis of semiconductor and hybrid metal/semiconductor nanomaterials can be used for biomedical and environmental applications.

## Figures and Tables

**Figure 1 microorganisms-11-01810-f001:**
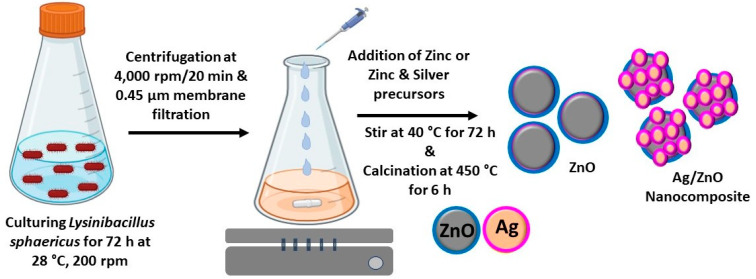
Schematic diagram of the biosynthesis of ZnO and Ag/ZnO NCs using the CFF of *L. sphaericus*.

**Figure 2 microorganisms-11-01810-f002:**
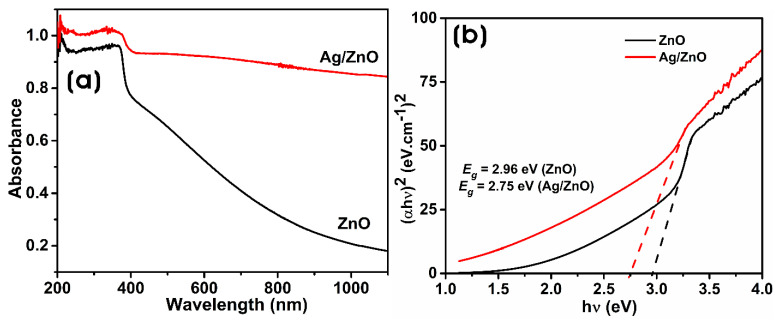
(**a**) UV-vis DRS spectra and (**b**) corresponding Tauc plots of biosynthesized ZnO and Ag/ZnO.

**Figure 3 microorganisms-11-01810-f003:**
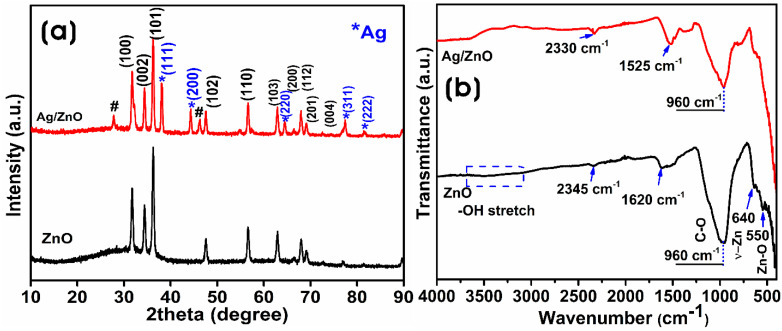
(**a**) XRD pattern (# peaks from macromolecules of the CFF) and (**b**) FTIR spectra of biosynthesized ZnO and Ag/ZnO.

**Figure 4 microorganisms-11-01810-f004:**
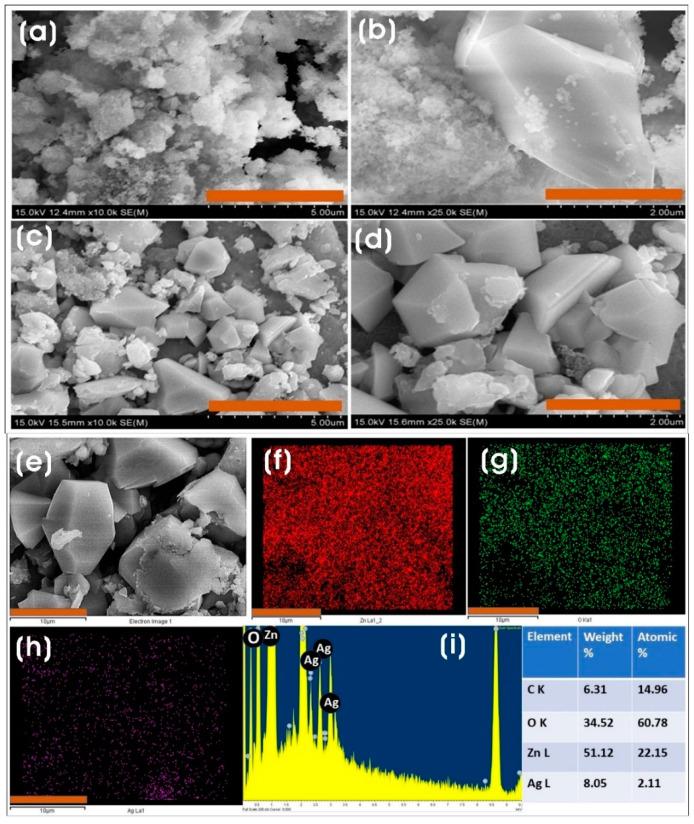
FE-SEM images of (**a**,**b**) ZnO and (**c**,**d**) Ag/ZnO at different magnifications. Scale bars: (**a**,**c**) 5 µm, and (**b**,**d**) 2 µm. (**e**) FE-SEM-EDX micrograph of Ag/ZnO and elemental maps of (**f**) Zn-L, (**g**) O-K, (**h**) Ag-L, and (**i**) EDX spectrum and the table with elemental composition. Scale bars, (**e**–**h**) 10 µm.

**Figure 5 microorganisms-11-01810-f005:**
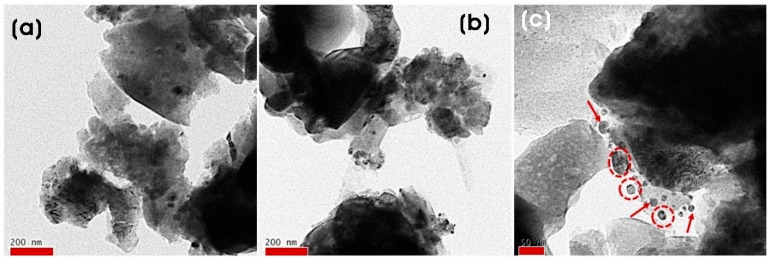
FE-TEM images of biosynthesized Ag/ZnO NCs at different magnifications. Arrows indicate the formation of Ag NPs on the surface of ZnO. Scale bars: (**a**,**b**) 200 nm and (**c**) 50 nm.

**Figure 6 microorganisms-11-01810-f006:**
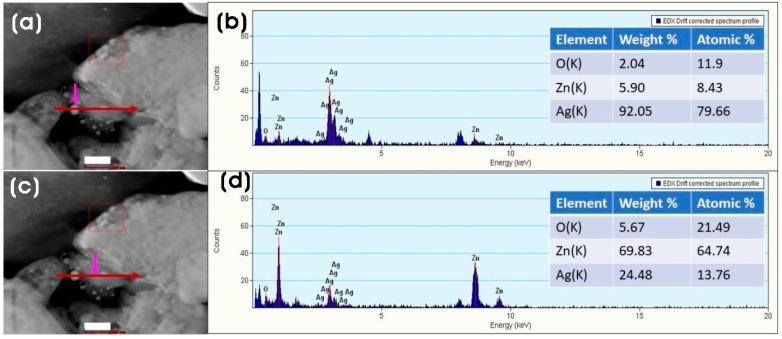
(**a**,**c**) HAADF-STEM images of Ag/ZnO and corresponding; (**b**,**d**) EDX line scan maps of the arrow-indicated areas of (**a**) Ag and (**c**) ZnO (inset shows the table of elemental composition). Scale bar: (**a**,**c**) 100 nm.

**Figure 7 microorganisms-11-01810-f007:**
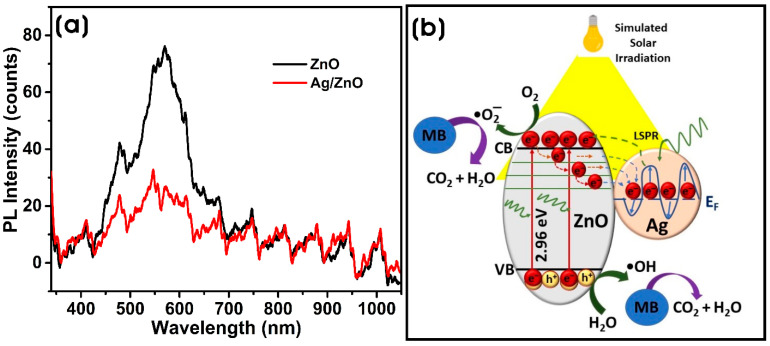
(**a**) Photoluminescence (PL) spectra of biosynthesized ZnO and Ag/ZnO NCs, (**b**) Schematic illustration of the probable photodegradation mechanism of methylene blue (MB) by biosynthesized Ag/ZnO.

**Figure 8 microorganisms-11-01810-f008:**
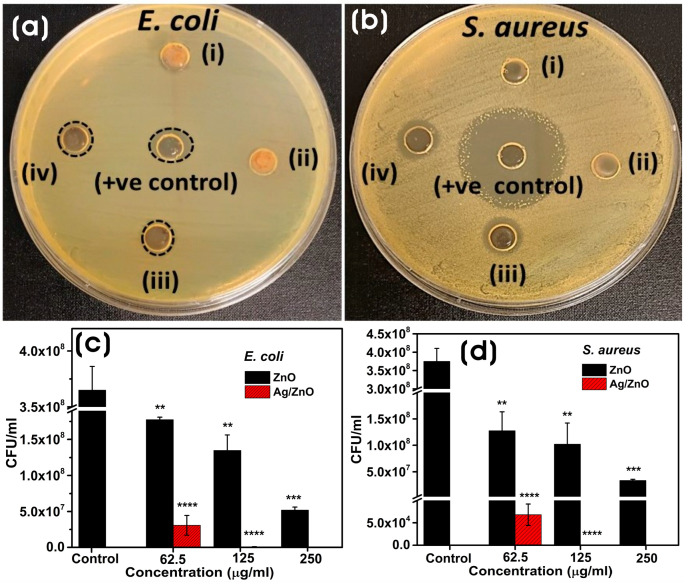
Antibacterial activity of biosynthesized ZnO and Ag/ZnO NCs against *E. coli* and *S. aureus*. (**a**,**b**) agar-well diffusion assay with (i) ZnO (200 µg), (ii) ZnO (400 µg), (iii) Ag/ZnO (200 µg) and (iv) Ag/ZnO (400 µg), and (**c**,**d**) bacterial colony counting assay at different concentrations. The results are the means ± SD of three replicates. The significance of the difference between control and bacteria treated with different concentrations of nanocomposites was determined by a two-tailed unpaired Welch’s *t*-test (** *p* ≤ 0.01; *** *p* ≤ 0.001; **** *p* ≤ 0.0001).

**Figure 9 microorganisms-11-01810-f009:**
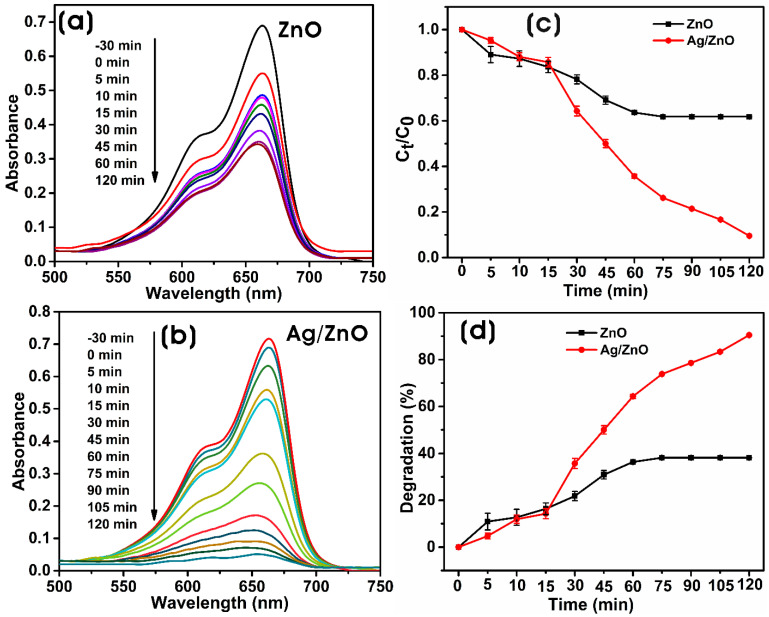
UV-vis absorbance spectra of the photodegradation of MB by biosynthesized (**a**) ZnO and (**b**) Ag/ZnO. (**c**) The plot of C_t_/C_0_ versus time and (**d**) the degradation percentage of MB by biosynthesized ZnO and Ag/ZnO.

**Figure 10 microorganisms-11-01810-f010:**
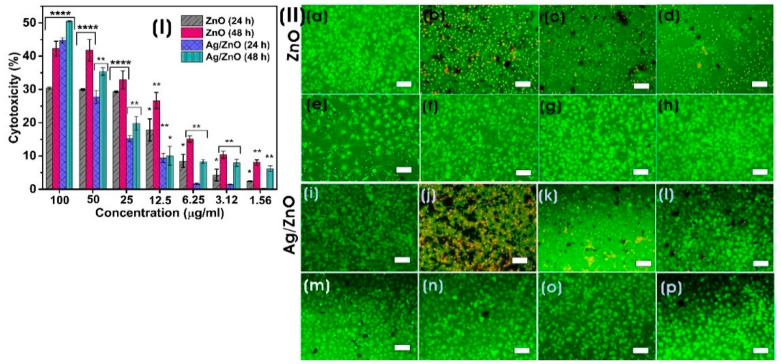
(**I**) In vitro cytotoxicity of biosynthesized ZnO and Ag/ZnO NCs on human cervical cancer HeLa cells for 24 and 48 h. Data are expressed as mean ± SD (three replicates). The significance of the difference between the cytotoxicity of different concentrations of NPs with control at 24 and 48 h was determined by a two-tailed unpaired Welch’s *t*-test (* *p* ≤ 0.05; ** *p* ≤ 0.01; **** *p* ≤ 0.0001). (**II**) Live/dead staining assay of HeLa cancer cells on exposure to biosynthesized ZnO and Ag/ZnO after 48 h (scale bar: 100 µm). (**a**,**i**) control, (**b**,**j**) 100 µg/mL, (**c**,**k**) 50 µg/mL, (**d**,**l**) 25 µg/mL, (**e**,**m**) 12.5 µg/mL, (**f**,**n**) 6.25 µg/mL, (**g**,**o**) 3.12 µg/mL, and (**h**,**p**) 1.56 µg/mL of ZnO and Ag/ZnO, respectively.

## Data Availability

Data is available under request.

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
