# Peer review of "Photocatalytic Degradation, Anticancer, and Antibacterial Studies of Lysinibacillus sphaericus Biosynthesized Hybrid Metal/Semiconductor Nanocomposites"

_microorganisms, 2023, doi:10.3390/microorganisms11071810_

Round 1
Reviewer 1 Report
The manuscript by Kannan Badri Narayanan et al. entitled ‘Photocatalytic degradation, anticancer, and antibacterial studies of Lysinibacillus sphaericus biosynthesized semiconductor and hybrid metal/semiconductor nanoparticles’ describes a study showcasing the possibility of using Lysinibacillus sphaericus cell-free filtrate for the preparation of zinc oxide based materials. The ZnO and Ag-doped ZnO were then assessed for their capability as photocatalysts in methylene blue degradation, their anticancer activity, and antibacterial activity.
The biological methods for preparation of materials are gaining importance for their ecofriendliness and possibility of surface modification of the prepared particles with biomolecules, so I find the subject important. The manuscript fits the scope of the journal ‘Microorganisms’.
There are however some issues that have to be addressed:
1. I am not convinced that the prepared materials are in fact nanoparticles. You mention it many times, but the results show a rather larger scale of the composites. The SEM and TEM micrographs are inconclusive, as they are too small to interpret them properly. The DLS results show clearly the hydrodynamic diameters above 1000 nm. In the manuscript no size given is in the nano scale. Please confirm the particle size.
2. In the mechanism of photodegradation you mention mineralization of methylene blue, it is also showcased in the Figure 5b. However, you never confirmed it. For such statement additional test should be performed, i.e. total organic carbon analysis. Methylene blue may be degraded to smaller molecules not absorbing light in the red region of the spectrum. Also, the C0 should be the concentration of MB after the dark period of adsorption-desorption. I did not find the information whether the MB solution was irradiated without the presence of the photocatalyst – this could also impact the results, as MB undergoes photobleaching by itself as well. As a result, the ‘Degradation’ as shown on the y axis in Figure 6d should be changed to ‘Removal’. Additionally, this part could also use some toxicity assessment of the post-degradation solution.
3. The 3.3.1.1 section is based on the literature only. You assume that the mechanism is what you described but additional experiments should be performed to confirm it – for example EPR experiments or photodegradation experiments with ROS quenchers and how it affects the photodegradation efficiency.
4. The anticancer part – to consider these materials as anticancer agents, experiments with healthy (non-cancerous) cells should be provided. Also, the big size of the particles (>1000 nm in DLS) clearly show the poor prognosis of potential clinical application – particles of such huge size would not reach the target tissues/cells.
5. Please describe in the Figure caption the regions highlighted in Figure 2d, as it is unclear without reading the whole article.
6. There seems to be a discrepancy in the description of the scale bars in the Figure 3 caption and in the scale bars present in the figure itself.
7. What are the unmarked reflections at ~28 and 35° in the XRPD?
8. It is weird to me that you mention that ‘various physical and chemical methods (…) are expensive and require sophisticated instruments’ and at the same time rely on photodegradation. This seems contradictory. Also, Fenton process and photodegradation are types of AOPs and here it seems that they are separate methods.
9. Two authors have no affiliations: Rakesh Bhaskar and Yong Joo Seok. There are also some of the CReDiT statements missing, most notably investigation.
Author Response
Reviewer #1
Comments and Suggestions for Authors
The manuscript by Kannan Badri Narayanan et al. entitled ‘Photocatalytic degradation, anticancer, and antibacterial studies of Lysinibacillus sphaericus biosynthesized semiconductor and hybrid metal/semiconductor nanoparticles’ describes a study showcasing the possibility of using Lysinibacillus sphaericus cell-free filtrate for the preparation of zinc oxide-based materials. The ZnO and Ag-doped ZnO were then assessed for their capability as photocatalysts in methylene blue degradation, their anticancer activity, and antibacterial activity.
The biological methods for preparation of materials are gaining importance for their ecofriendliness and possibility of surface modification of the prepared particles with biomolecules, so I find the subject important. The manuscript fits the scope of the journal ‘Microorganisms’.
There are however some issues that have to be addressed:
- I am not convinced that the prepared materials are in fact nanoparticles. You mention it many times, but the results show a rather larger scale of the composites. The SEM and TEM micrographs are inconclusive, as they are too small to interpret properly. The DLS results clearly show hydrodynamic diameters above 1000 nm. In the manuscript no size given is in the nano scale. Please confirm the particle size.
Response: Yes, thank you. We have changed the term “nanoparticles” to nanocomposite for Ag/ZnO. DLS and SEM results can be influenced by aggregation and particle stability. Moreover, DLS results differ with changes in the morphology of particles. XRD patterns and TEM micrographs indicate the embedment of nanocrystalline silver on the surface of ZnO submicronic particles. TEM images show the presence of Ag NPs with a size of 19.0 ± 11.9 nm attached to the irregularly shaped, puffy-like ZnO particles (Page No. 18; Line No. 6-7). The crystallite sizes of ZnO and Ag NPs in Ag/ZnO calculated using the Debye-Scherrer equation were 25.6 ± 9.9 nm and 24.9 ± 1.9 nm, respectively (Page No. 15; Line No. 1). We also made figure 4 into figure 4 and figure 5 for better understanding.
- In Figure of photodegradation you mention mineralization of methylene blue, it is also showcased in Figure 5b. However, you never confirmed it. For such a statement an additional test should be performed, i.e., total organic carbon analysis. Methylene blue may be degraded to smaller molecules not absorbing light in the red region of the spectrum. Also, the C0should be the concentration of MB after the dark period of adsorption-desorption. I did not find the information whether the MB solution was irradiated without the presence of the photocatalyst – this could also impact the results, as MB undergoes photobleaching by itself as well. As a result, the ‘Degradation’ as shown on the y axis in Figure 6d should be changed to ‘Removal’. Additionally, this part could also use some toxicity assessment of the post-degradation solution.
Response: We performed only elemental analysis of ZnO and Ag/ZnO, and the results of the weight% and atomic% of carbon content were estimated through SEM-EDX and TEM-EDX analysis. In the presence of a catalyst, the cationic thiazine dye, MB was reduced into less toxic colorless leuco-methylene blue (LMB). Many previous reports explain that MB gets degraded into LMB with a decrease in the peak at 295 nm. We also performed the degradation of MB without catalyst, which showed that the MB concentration was slowly reduced by photoreaction i.e., degradation of 25.8 ± 1.91% after 180 min (Supplementary Figure 3) (Page No. 24, Line No.3-5; 14-18). We are planning to perform a full-fledged phototoxicity analysis with various degradation analyses as our next project.
Supplementary Figure 3. Photolysis of methylene blue dye without any photocatalysts.
- The 3.3.1.1 section is based on literature only. You assume that the mechanism is what you described but additional experiments should be performed to confirm it – for example EPR experiments or photodegradation experiments with ROS quenchers and how it affects the photodegradation efficiency.
Response: Thank you. We removed section 3.3.1.1. and merged it with 3.3.1. section. Yes, this is the proposed mechanism of degradation, and it has been already proven by many researchers with similar silver/zinc oxide nanocomposites [65-67]. The importance of this composite is its biological synthesis and established the result of having the potency to involve in photodegradation, antimicrobial, and anticancer results.
- The anticancer part – to consider these materials as anticancer agents, experiments with healthy (non-cancerous) cells should be provided. Also, the big size of the particles (>1000 nm in DLS) clearly show the poor prognosis of potential clinical application – particles of such huge size would not reach the target tissues/cells.
Response: DLS results are the qualitative result, they vary with changes in the morphology of the nanoparticles/nanocomposites. TEM is the only possible confirmatory experiment to confirm the size of nanoparticles. TEM shows the formation of Ag NPs with a size of 19.0 ± 11.9 nm attached to the submicronic, irregularly shaped, puffy-like ZnO particles. Nanoparticles with a diameter of 10-100 nm are considered efficient anticancer agents due to their effective delivery and enhanced permeability and retention (EPR) effects. Smaller particles are considered to undergo rapid release from normal vessels and damage healthy cells and tissue before their metabolism by the kidneys (<10 nm in size) [68]. Moreover, anticancer activity can be exhibited by local administration on the targeted site for photodynamic therapy. These composite materials act as a reservoir of releasing metal ions inducing oxidative stress required for the inhibition of the cancer cell's growth and death (Page No. 27, Line No. 10-13)
- Please describe in the Figure caption the regions highlighted in Figure 2d, as it is unclear without reading the whole article.
Response: Thank you, we have elaborated on the peaks in figure 2d (Figure 3b).
- There seems to be a discrepancy in the description of the scale bars in the Figure 3 caption and in the scale, bars present in the figure itself.
Response: Thank you. Yes, we have corrected the scale bars for Figure 3 (Figure 4).
- What are the unmarked reflections at ~28 and 35° in the XRPD?
Response: XRD peaks at 28 and 45 are probably from the macromolecules present in the CFF (Page No. 14, Line No. 2).
- It is weird to me that you mention that ‘various physical and chemical methods (…) are expensive and require sophisticated instruments’ and at the same time rely on photodegradation. This seems contradictory. Also, the Fenton process and photodegradation are types of AOPs and here it seems that they are separate methods.
Response: Thank you for mentioning the wrong statement. We have modified the statement for readers' clear understanding (Page No. 3, Line No. 16-22).
- Two authors have no affiliations: Rakesh Bhaskar and Yong Joo Seok. There are also some of the CReDiT statements missing, most notably investigation.
Response: Thank you for correctly pointing out the mistake. We have added the missing affiliation data for the authors.

Reviewer 2 Report
The paper's theme is about photocatalytic degradation, anticancer, and antibacterial studies using semiconductor and hybrid metal/semiconductor nanoparticles biosynthesized by Lysinibacillus sphaericus. These nanoparticles are composed of Zinc Oxide (ZnO) and Silver/Zinc Oxide (Ag/ZnO). The paper includes some experimental results, such as the results of antibacterial activity tests, and some discussion about the formation of nanoparticles. Nonetheless, the discussion surrounding the experimental results within this article is excessively simplistic, precluding the reader from gleaning any substantial information. Further compounding the issue, the article harbors numerous unresolved queries, reminiscent of an unfinished piece of work. Therefore, I would not recommend acceptance and publication of this manuscript until these issues are addressed and resolved.
While I have endeavored to identify as many areas for improvement as possible, this does not preclude the possibility of similar issues elsewhere in the manuscript. The identified issues include:
1. What prompts the selection of Zinc oxide as a choice among an array of photocatalysts? Could you elaborate on the rationale behind incorporating silver with zinc oxide in the synthesis of nanomaterials?
2. Could you elucidate on why the cell-free filtrate synthesis of Lactobacillus globulus is synthetic? What distinctive benefits can be derived from the filtrate of this particular cell?
3. The labels attached to the images require careful scrutiny, as Figure 2 seems to be duplicated.
4. How does Fourier Transform Infrared Spectroscopy (FTIR) complement the synthetic pathway in discerning which functional group within the filtrate is supplied by sugars or proteins and thus provides a site for synthesis?
5. What constitutes the basis for selecting E. coli and Staphylococcus aureus for bacteriostatic testing?
6. The clarity of the Scanning Electron Microscopy (SEM) image leaves room for improvement.
7. Could you elaborate on the correlation between the treatment of azo dye methylene blue and the antibacterial and antitumor properties of nanomaterials?
8. Could you provide a brief overview of the bottom-up synthesis method and how it intersects with bio-green synthesis? Was this particular method of synthesis utilized in your study?
9. Can the biosynthetic process be optimized to enhance the yield or purity of these nanoparticles?
10. How would you assess the efficacy of these nanoparticles in anticancer and antibacterial applications? Could modifications in their size, shape, or surface functionalization potentially enhance this efficiency? Additionally, what are your insights on the biocompatibility of these nanoparticles? Could alterations in their chemical composition or surface treatment potentially augment their biocompatibility?
11. Could you comment on the stability of these nanoparticles? Would it be plausible to improve stability by adjusting their storage conditions or incorporating stabilizers?
Author Response
Reviewers #2
The paper's theme is about photocatalytic degradation, anticancer, and antibacterial studies using semiconductor and hybrid metal/semiconductor nanoparticles biosynthesized by Lysinibacillus sphaericus. These nanoparticles are composed of Zinc Oxide (ZnO) and Silver/Zinc Oxide (Ag/ZnO). The paper includes some experimental results, such as the results of antibacterial activity tests, and some discussion about the formation of nanoparticles. Nonetheless, the discussion surrounding the experimental results within this article is excessively simplistic, precluding the reader from gleaning any substantial information. Further compounding the issue, the article harbors numerous unresolved queries, reminiscent of an unfinished piece of work. Therefore, I would not recommend acceptance and publication of this manuscript until these issues are addressed and resolved.
While I have endeavored to identify as many areas for improvement as possible, this does not preclude the possibility of similar issues elsewhere in the manuscript. The identified issues include:
- What prompts the selection of Zinc oxide as a choice among an array of photocatalysts? Could you elaborate on the rationale behind incorporating silver with zinc oxide in the synthesis of nanomaterials?
Response: Zinc oxide is cost-effective, non-toxic, and a more efficient photocatalyst than TiO2. Doping and recombination of ZnO with transition metals are used to improve the photoresponse range of ZnO as a photocatalyst in a wide wavelength range. This is the rationale for the synthesis of silver-zinc oxide nanocomposite (Page No. 4, Line No. 13-14, 18-20).
- Could you elucidate why the cell-free filtrate synthesis of Lactobacillus globulus is synthetic? What distinctive benefits can be derived from the filtrate of this particular cell?
Response: Bacterial cell filtrate contains many oxidoreductases and other biomacromolecules, which are involved in the synthesis of ZnO and Ag NPs.
- The labels attached to the images require scrutiny, as Figure 2 seems to be duplicated.
Response: Thank you. We have modified the figure labels of Figure 2, 3 and other figures.
- How does Fourier Transform Infrared Spectroscopy (FTIR) complement the synthetic pathway in discerning which functional group within the filtrate is supplied by sugars or proteins and thus provides a site for synthesis?
Response: Biological macromolecules are rich in carbon, hydrogen, and oxygen. The functional groups related to these produce bands at 960 cm–1 was correlated to the vibrations of C–O bonds. The calcination process burns off the template and the functional groups of the stabilizing moieties from the cell-free filtrate. Thus, a very less significant stabilizing agent caps the nanoparticles and nanocomposites.
- What constitutes the basis for selecting E. coli and Staphylococcus aureus for bacteriostatic testing?
Response: Antibacterial activity of ZnO and Ag/ZnO were evaluated against Gram-negative (E. coli) and Gram-positive (S. aureus) as test bacteria, as all pathogenic bacteria come in these two classifications.
- The clarity of the Scanning Electron Microscopy (SEM) image leaves room for improvement.
Response: Thank you. Yes, we have improved Figure 3 (Figure 4) for better understanding.
- Could you elaborate on the correlation between the treatment of azo dye methylene blue and the antibacterial and antitumor properties of nanomaterials?
Response: Our nanomaterial has versatile applications capability; it can be used both in environmental and biomedical applications.
- Could you provide a brief overview of the bottom-up synthesis method and how it intersects with bio-green synthesis? Was this method of synthesis utilized in your study?
Response: Yes, we have incorporated a paragraph related to bottom-up synthesis and its importance in green synthesis. We have utilized such a bottom-up approach using a microbial methodology (Page No.10, Line No.12-15; Page No. 11, Line No. 1-2).
- Can the biosynthetic process be optimized to enhance the yield or purity of these nanoparticles?
Response: Thank you. Before experimenting, preliminary studies of optimization of higher yield of ZnO and Ag/ZnO were performed. Our data is the exhibition of optimized results.
- How would you assess the efficacy of these nanoparticles in anticancer and antibacterial applications? Could modifications in their size, shape, or surface functionalization potentially enhance this efficiency? Additionally, what are your insights on the biocompatibility of these nanoparticles? Could alterations in their chemical composition or surface treatment potentially augment their biocompatibility?
Response: Biocompatibility comes from the inertness of the components used in the nanoparticles, but some metal nanoparticles produce cytotoxicity through oxidative stress on cells, leading to cell damage and death. Both these properties are related to their size, shape, and surface functionalization.
- Could you comment on the stability of these nanoparticles? Would it be plausible to improve stability by adjusting their storage conditions or incorporating stabilizers?
Response: Zeta potential values demonstrate the stability of synthesized ZnO and Ag/ZnO. Supplementary Figure 1 shows that ZnO and Ag/ZnO have -30.1 ± 8.39 and -29 ± 5.74 mV, respectively. These values indicate that nanoparticles are moderately stable (Page No. 16, Line No. 1-6).

Reviewer 3 Report
This is a good and very interesting work in the field of biosynthesis, green chemistry and ecological chemistry, this is a very important trend in the development of modern science. This article needs to be published. I propose a revision to improve the manuscript. And so, no doubt, the article is very good and will be well cited.
The abstract should be shortened, now it's very boring
In vitro and in vivo should be written in italics everywhere
Everywhere should be the same font and size
What is the degree of transformation of zinc nitrate into zinc hydroxide? Why was zinc oxide formed and not Zn(OH)2? Does the reaction mixture contain zinc complexes?
Author Response
Reviewer #3
This is a good and very interesting work in the field of biosynthesis, green chemistry, and ecological chemistry. This is a very important trend in the development of modern science. This article needs to be published. I propose a revision to improve the manuscript. And so, no doubt, the article is very good and will be well cited.
The abstract should be shortened, now it's very boring.
Response: Thank you. The abstract is 225 words, we have modified the abstract without losing the significance and novelties of the work.
In vitro and in vivo should be written in italics everywhere.
Response: Yes, we have italicized in vitro and in vivo words (Page No. 5, Line No. 13; Page No. 8, Line No. 17; Page No. 9, Line No. 2, 10; Page No. 26, Line No. 17; Page No. 27, Line No. 1).
Everywhere should be the same font and size.
Response: Thank you. We have thoroughly checked the font and size.
What is the degree of transformation of zinc nitrate into zinc hydroxide? Why was zinc oxide formed and not Zn(OH)2? Does the reaction mixture contain zinc complexes?
Response: Zinc nitrate (Zn(NO3)2) reacts with the hydroxide group forming zinc hydroxide (Zn(OH)2), which decomposes to form zinc oxide (ZnO) under calcination.

Reviewer 4 Report
Dear authors,
Thank you for your submission.
This is an interesting study. However, its presentation is not acceptable. I recommend some amendments.
1. The title should be restructured to conform to the aim of the study.
2. Long and incomplete sentences should be avoided in the Abstract
3. In the introduction, a one-sided discussion of the textile field should be avoided and it is better to discuss a broad field with references.
4. “Metal/metal oxide nanoparticles composed of gold, silver, zinc, copper, palladium, platinum, or ruthenium are commonly used photocatalysts for environmental remediation applications.” Please add references for each metal.
5. I think it would be better if the authors wrote synthesis and characterization, then bacterial culture, and then followed cytotoxicity. The order of results and discussions as materials and methods should be maintained.
6. Figure 2 is randomly arranged. Please arrange it consistently.
7. In Figure 3, the scale bars were not dealt with well. The image quality is not good.
8. The properties of nanoparticles are not clearly defined. It should be well-identified with scale bars and measurement lines/dots. TEM images are not well-defined.
9. XRD peaks should be discussed with references.
10. The mechanism of photocatalytic degradation is unclear.
11. A brief conclusion should appear with the consistency of the study's aim and results. Exaggeration should be avoided.
12. Supplementary data should be well cited in the main text and a brief should be discussed in the text or figure legends in the supplementary text file.
Please improve the manuscripts. I look forward to seeing the revised manuscript.
Thank you
Long and wordy sentences should be avoided.
Incomplete sentences should be avoided.
Author Response
Reviewer #4
This is an interesting study. However, its presentation is not acceptable. I recommend some amendments.
- The title should be restructured to conform to the aim of the study.
Response: We have changed the title to “Photocatalytic degradation, anticancer, and antibacterial studies of Lysinibacillus sphaericus biosynthesized hybrid metal/semiconductor nanocomposites”
- Long and incomplete sentences should be avoided in the Abstract
Response: Thank you. We have modified the abstract.
- In the introduction, a one-sided discussion of the textile field should be avoided, and it is better to discuss a broad field with references.
Response: As suggested, we have included topics of anticancer and antimicrobial in the introduction (Page No. 5; Line No. 1-10).
- “Metal/metal oxide nanoparticles composed of gold, silver, zinc, copper, palladium, platinum, or ruthenium are commonly used photocatalysts for environmental remediation applications.” Please add references for each metal.
Response: Yes, we have added references for the above sentence (Page No.4, Line No.11-12).
- I think it would be better if the authors wrote synthesis and characterization, then bacterial culture, and then followed cytotoxicity. The order of results and discussions as materials and methods should be maintained.
Response: Thank you. As suggested, we have rearranged the order of results and discussion in the order of materials and methods.
- Figure 2 is randomly arranged. Please arrange it consistently.
Response: Thank you. We have arranged Figure 2 into two figures: Figure 2 and Figure 3.
- In Figure 3, the scale bars were not dealt with well. The image quality is not good.
Response: Thank you. We have changed the scale bar, figure legends, and the arrangement of Figure 3 (Figure 4) for clear understanding.
- The properties of nanoparticles are not clearly defined. It should be well-identified with scale bars and measurements lines/dots. TEM images are not well-defined.
Response: Figure 4 has been made into separate Figures 5 and 6 with a well-defined scale bar for clear understanding. The presence of Ag NPs on ZnO was well marked with red circles.
- XRD peaks should be discussed with references.
Response: Yes, we have added references for XRD results (Page No.14, Line No. 9).
- The mechanism of photocatalytic degradation is unclear.
Response: Thank you. We have rewritten the mechanism of photocatalytic degradation for clear understanding (Page No.25, Line No. 19-23; Page No. 26, Line No. 1-14).
- A brief conclusion should appear with the consistency of the study's aim and results. Exaggeration should be avoided.
Response: Thank you. The conclusion section has been well-modified to explain the results.
- Supplementary data should be well cited in the main text and a brief should be discussed in the text or figure legends in the supplementary text file.
Response: Yes, we have cited supplementary figures in the text (Suppl Fig. 1 (Page No.16, Line No. 6,) Suppl Fig. 2 (Page No.17, Line No.17), Suppl Fig. 3 (Page No.24, Line No.16). Supplementary figures along with figure captions are provided in the supplementary materials file.
- Please improve the manuscripts. I look forward to seeing the revised manuscript.
Response: Thank you for your valuable suggestions. We have improved the manuscript as mentioned.

Reviewer 5 Report
There are still some minor revisions that I suggest for its publication as seen below.
1) Correct the use of abbreviation i.e., cell-free filtrate (CFF) etc. at different places.
2) Include a schematic diagram of the biosynthesis of ZnO (semiconductor) and Ag/ZnO (metal/semiconductor) nanoparticles process in section 2.2 Biosynthesis of semiconductor and metal/semiconductor nanoparticles.
3) There is some clumsy English grammar/phrasing throughout the manuscript. English/ grammar of the manuscript can be improved before publication.

N/A
Author Response
Reviewer #5
There are still some minor revisions that I suggest for its publication as seen below.
1) Correct the use of abbreviation i.e., cell-free filtrate (CFF) etc. at different places.
Response: Thank you. We have mentioned the expansion only for the first time and used abbreviations throughout the manuscript.
2) Include a schematic diagram of the biosynthesis of ZnO (semiconductor) and Ag/ZnO (metal/semiconductor) nanoparticles process in section 2.2 Biosynthesis of semiconductor and metal/semiconductor nanoparticles.
Response: Yes, as suggested, we have mentioned the schematic diagram of the biosynthesis of ZnO and Ag/ZnO in Section 2.2. (Page No. 7; Line No. 5).
3) There is some clumsy English grammar/phrasing throughout the manuscript. English/ grammar of the manuscript can be improved before publication.
Response: Thank you. We have thoroughly edited the manuscript with a Grammar editing service and highlighted it in red color.

Reviewer 6 Report
Upon reviewing your manuscript intitled: Photocatalytic degradation, anticancer, and antibacterial studies of Lysinibacillus sphaericus biosynthesized semiconductor and hybrid metal/semiconductor nanoparticles. I find your work interesting, but I do not believe it can be published in its current form. Therefore, some revisions must be done so that it may be published in this journal. I have the following comments and recommendations.
- The summary is very qualitative. Does not present the main novelties of the study
- The introduction of relevant background and research progress was not comprehensive enough. Several recent works that address the theme of this proposal were not mentioned. In this sense, the references must be updated, to highlight the topicality of the subject studied.
- The doping can improve several material properties and the choice for doping depends on several factors. With respect to this, the authors do not justify the doping concentrations used. Due to the importance for this system, a reasoned explanation must be attached as part of the motivation and justification of the work.
- Include the purity of the raw materials used.
- An unidentified phase is present in the XRD (Ag/ZnO), identify and comment
- The crystallite sizes for the various samples were calculated using the Debye-Scherrer equation. However, it is well known that the peak broadening is impacted by other factors such as instrument-related broadening, residual stresses in crystals, etc. How did the authors determine or subtract these well-known parameters?
- - There is no experimental data on chemical stability of synthesized materials
- As an environmental restoration material, recycling must be considered, and some tests and analysis should be supplemented.
- Why the authors selected Methylene Blue (MB) and rhodamine B for the degradation study? When there are lot of reports are available in the literature.
Author Response
Reviewer #6
Upon reviewing your manuscript entitled: Photocatalytic degradation, anticancer, and antibacterial studies of Lysinibacillus sphaericus biosynthesized semiconductor and hybrid metal/semiconductor nanoparticles. I find your work interesting, but I do not believe it can be published in its current form. Therefore, some revisions must be done so that it may be published in this journal. I have the following comments and recommendations.
- The summary is very qualitative. Does not present the main novelties of the study.
Response: Yes, we have included the significance of the study in the summary and highlighted in red color.
- The introduction of relevant background and research progress was not comprehensive enough. Several recent works that address the theme of this proposal were not mentioned. In this sense, the references must be updated, to highlight the topicality of the subject studied.
Response: Thank you, we have modified the introduction with additional topics and also updated the references section (Page No. 3, Line No. 16-22; Page No. 5, Line No. 1-10).
- Doping can improve several material properties and the choice for doping depends on several factors. With respect to this, the authors do not justify the doping concentrations used. Due to the importance for this system, a reasoned explanation must be attached as part of the motivation and justification of the work.
Response: The dopant concentration was based on previous literature and our work. In our previous work, we used 0.2%, 0.4%, and 0.8% (w/v) silver as a dopant, so we want to try around this concentration to check the efficiency of the Ag/ZnO nanocomposite (Sharwani et al. 2022) (reference 1).
- Include the purity of the raw materials used.
Response: Raw materials used were 99% purity and mentioned in section 2. Materials (Page No. 5, Line No.18; Page No. 6, Line No. 1).
- An unidentified phase is present in the XRD (Ag/ZnO), identify and comment.
Response: XRD peaks at 28 and 45 are probably from the macromolecules present in the CFF (Page No. 14, Line No. 2).
- The crystallite sizes for the various samples were calculated using the Debye-Scherrer equation. However, it is well known that the peak broadening is impacted by other factors such as instrument-related broadening, residual stresses in crystals, etc. How did the authors determine or subtract these well-known parameters?
Response: Thank you. We just used the Debye-Scherrer equation to calculate the crystallite size. We did not perform any deduction of well-known influencing factors.
- There is no experimental data on chemical stability of synthesized materials.
Response: Yes, we did not perform any experiments to check the stability of materials. But, we performed zeta potential analysis, which shows that both ZnO and Ag/ZnO possess around - 30 mV, which indicates that they are moderately stable in aqueous solution (Supplementary Figure 1; Page No. 16; Line No. 2-6).
- As an environmental restoration material, recycling must be considered, and some tests and analysis should be supplemented.
Response: As suggested, we have performed the reusability of Ag/ZnO, and the data is amended in the results and discussion section (Page No. 25, Line No. 9-17) (Supplementary Figure 4).
- Why the authors selected Methylene Blue (MB) and rhodamine B for the degradation study? When there are lot of reports available in the literature.
Response: This is the first report on the synthesis of Ag/ZnO using the bacterial cell-free filtrate, so we need to evaluate their potency in their potency in photodegradation, antimicrobial and anticancer activities.

Round 2
Reviewer 1 Report
Reviewer #1
Comments and Suggestions for Authors
The manuscript by Kannan Badri Narayanan et al. entitled ‘Photocatalytic degradation, anticancer, and antibacterial studies of Lysinibacillus sphaericus biosynthesized semiconductor and hybrid metal/semiconductor nanoparticles’ describes a study showcasing the possibility of using Lysinibacillus sphaericus cell-free filtrate for the preparation of zinc oxide-based materials. The ZnO and Ag-doped ZnO were then assessed for their capability as photocatalysts in methylene blue degradation, their anticancer activity, and antibacterial activity.
The biological methods for preparation of materials are gaining importance for their ecofriendliness and possibility of surface modification of the prepared particles with biomolecules, so I find the subject important. The manuscript fits the scope of the journal ‘Microorganisms’.
There are however some issues that have to be addressed:
1. I am not convinced that the prepared materials are in fact nanoparticles. You mention it many times, but the results show a rather larger scale of the composites. The SEM and TEM micrographs are inconclusive, as they are too small to interpret properly. The DLS results clearly show hydrodynamic diameters above 1000 nm. In the manuscript no size given is in the nano scale. Please confirm the particle size.
Response: Yes, thank you. We have changed the term “nanoparticles” to nanocomposite for Ag/ZnO. DLS and SEM results can be influenced by aggregation and particle stability. Moreover, DLS results differ with changes in the morphology of particles. XRD patterns and TEM micrographs indicate the embedment of nanocrystalline silver on the surface of ZnO submicronic particles. TEM images show the presence of Ag NPs with a size of 19.0 ± 11.9 nm attached to the irregularly shaped, puffy-like ZnO particles (Page No. 18; Line No. 6-7). The crystallite sizes of ZnO and Ag NPs in Ag/ZnO calculated using the Debye-Scherrer equation were 25.6 ± 9.9 nm and 24.9 ± 1.9 nm, respectively (Page No. 15; Line No. 1). We also made figure 4 into figure 4 and figure 5 for better understanding.
[Second review] The term nanocomposite is not appropriate as well. Only the silver nanoparticles in the composite materials are proven to be of nanometric dimensions. The calculated crystallite size does not reflect the actual particle size.
2. In Figure of photodegradation you mention mineralization of methylene blue, it is also showcased in Figure 5b. However, you never confirmed it. For such a statement an additional test should be performed, i.e., total organic carbon analysis. Methylene blue may be degraded to smaller molecules not absorbing light in the red region of the spectrum. Also, the C0 should be the concentration of MB after the dark period of adsorption-desorption. I did not find the information whether the MB solution was irradiated without the presence of the photocatalyst – this could also impact the results, as MB undergoes photobleaching by itself as well. As a result, the ‘Degradation’ as shown on the y axis in Figure 6d should be changed to ‘Removal’. Additionally, this part could also use some toxicity assessment of the post-degradation solution.
Response: We performed only elemental analysis of ZnO and Ag/ZnO, and the results of the weight% and atomic% of carbon content were estimated through SEM-EDX and TEM-EDX analysis. In the presence of a catalyst, the cationic thiazine dye, MB was reduced into less toxic colorless leuco-methylene blue (LMB). Many previous reports explain that MB gets degraded into LMB with a decrease in the peak at 295 nm. We also performed the degradation of MB without catalyst, which showed that the MB concentration was slowly reduced by photoreaction i.e., degradation of 25.8 ± 1.91% after 180 min (Supplementary Figure 3) (Page No. 24, Line No.3-5; 14-18). We are planning to perform a full-fledged phototoxicity analysis with various degradation analyses as our next project.

Supplementary Figure 3. Photolysis of methylene blue dye without any photocatalysts.
[Second review] Thank you for the clarification. Based on the information given I am not sure whether the biosynthesized ZnO is photoactive. As the photodecomposition of MB alone is around 25% and the adsorption is responsible for around 20% decrease in MB concentration, This leaves about 5% that is photodegraded, as the Ct/C0 after 120 min is ~50%.
The description of the figure and in the main text also was not changed from photodegradation to removal which is more appropriate.
Also, as the LMB was not confirmed only assumed, I would change “was reduced into the less toxic colorless leuco-methylene blue (LMB)” to “probably was transformed into the less toxic colorless leuco-methylene blue (LMB)”.
3. The 3.3.1.1 section is based on literature only. You assume that the mechanism is what you described but additional experiments should be performed to confirm it – for example EPR experiments or photodegradation experiments with ROS quenchers and how it affects the photodegradation efficiency.
Response: Thank you. We removed section 3.3.1.1. and merged it with 3.3.1. section. Yes, this is the proposed mechanism of degradation, and it has been already proven by many researchers with similar silver/zinc oxide nanocomposites [65-67]. The importance of this composite is its biological synthesis and established the result of having the potency to involve in photodegradation, antimicrobial, and anticancer results.
[Second review] If this is the case, that the mechanism is well established, I do not see the need to present the whole mechanism in the paper. On the other hand – can you be sure that the use of CFF does not alter the mechanism of the photodegradation and ROS involved?
I do not understand the calcination of the material between the cycles – it is a rather non-standard procedure, as a photocatalyst should be effective without such treatment between cycles. Can you cite any references where such treatment was performed?
4. The anticancer part – to consider these materials as anticancer agents, experiments with healthy (non-cancerous) cells should be provided. Also, the big size of the particles (>1000 nm in DLS) clearly show the poor prognosis of potential clinical application – particles of such huge size would not reach the target tissues/cells.
Response: DLS results are the qualitative result, they vary with changes in the morphology of the nanoparticles/nanocomposites. TEM is the only possible confirmatory experiment to confirm the size of nanoparticles. TEM shows the formation of Ag NPs with a size of 19.0 ± 11.9 nm attached to the submicronic, irregularly shaped, puffy-like ZnO particles. Nanoparticles with a diameter of 10-100 nm are considered efficient anticancer agents due to their effective delivery and enhanced permeability and retention (EPR) effects. Smaller particles are considered to undergo rapid release from normal vessels and damage healthy cells and tissue before their metabolism by the kidneys (<10 nm in size) [68]. Moreover, anticancer activity can be exhibited by local administration on the targeted site for photodynamic therapy. These composite materials act as a reservoir of releasing metal ions inducing oxidative stress required for the inhibition of the cancer cell's growth and death (Page No. 27, Line No. 10-13)
[Second review] The particle size of 19 nm refers to silver, which is deposited on the surface of ZnO, which are considerably larger as seen in the micrographs and the only values of the particle size are the results from DLS. The DLS results are affected by a series of factors, however they show how big can the particles/agglomerates be regarded in aqueous medium (hydrodynamic diameter). SEM and TEM only show the actual particle size without the effect of the solvent or agglomeration. And particles >1000 nm, regardless whether these are individual particles or agglomerates, would trigger macrophage immune response and be quickly removed from bloodstream. If topical application is regarded such information should be added to the limitations of the study.
The healthy cells experiments would show that there are no leftover cellular elements present on the surface of the materials that would act in a cytotoxic manner. It should be performed.
5. Please describe in the Figure caption the regions highlighted in Figure 2d, as it is unclear without reading the whole article.
Response: Thank you, we have elaborated on the peaks in figure 2d (Figure 3b).
[Second review] Thank you for the changes made.
6. There seems to be a discrepancy in the description of the scale bars in the Figure 3 caption and in the scale, bars present in the figure itself.
Response: Thank you. Yes, we have corrected the scale bars for Figure 3 (Figure 4).
[Second review] Thank you for the changes made.
7. What are the unmarked reflections at ~28 and 35° in the XRPD?
Response: XRD peaks at 28 and 45 are probably from the macromolecules present in the CFF (Page No. 14, Line No. 2).
[Second review] Is it possible that the macromolecules were still present in their crystalline form after 450°C treatment?
8. It is weird to me that you mention that ‘various physical and chemical methods (…) are expensive and require sophisticated instruments’ and at the same time rely on photodegradation. This seems contradictory. Also, the Fenton process and photodegradation are types of AOPs and here it seems that they are separate methods.
Response: Thank you for mentioning the wrong statement. We have modified the statement for readers' clear understanding (Page No. 3, Line No. 16-22).
[Second review] Thank you for the changes made.
9. Two authors have no affiliations: Rakesh Bhaskar and Yong Joo Seok. There are also some of the CReDiT statements missing, most notably investigation.
Response: Thank you for correctly pointing out the mistake. We have added the missing affiliation data for the authors.
[Second review] Thank you for the changes made.
Author Response
Reviewer #1
Comments and Suggestions for Authors
The manuscript by Kannan Badri Narayanan et al. entitled ‘Photocatalytic degradation, anticancer, and antibacterial studies of Lysinibacillus sphaericus biosynthesized semiconductor and hybrid metal/semiconductor nanoparticles’ describes a study showcasing the possibility of using Lysinibacillus sphaericus cell-free filtrate for the preparation of zinc oxide-based materials. The ZnO and Ag-doped ZnO were then assessed for their capability as photocatalysts in methylene blue degradation, their anticancer activity, and antibacterial activity.
The biological methods for preparation of materials are gaining importance for their ecofriendliness and possibility of surface modification of the prepared particles with biomolecules, so I find the subject important. The manuscript fits the scope of the journal ‘Microorganisms’.
There are however some issues that have to be addressed:
- I am not convinced that the prepared materials are in fact nanoparticles. You mention it many times, but the results show a rather larger scale of the composites. The SEM and TEM micrographs are inconclusive, as they are too small to interpret properly. The DLS results clearly show hydrodynamic diameters above 1000 nm. In the manuscript no size given is in the nano scale. Please confirm the particle size.
Response (1): Yes, thank you. We have changed the term “nanoparticles” to nanocomposite for Ag/ZnO. DLS and SEM results can be influenced by aggregation and particle stability. Moreover, DLS results differ with changes in the morphology of particles. XRD patterns and TEM micrographs indicate the embedment of nanocrystalline silver on the surface of ZnO submicronic particles. TEM images show the presence of Ag NPs with a size of 19.0 ± 11.9 nm attached to the irregularly shaped, puffy-like ZnO particles (Page No. 18; Line No. 6-7). The crystallite sizes of ZnO and Ag NPs in Ag/ZnO calculated using the Debye-Scherrer equation were 25.6 ± 9.9 nm and 24.9 ± 1.9 nm, respectively (Page No. 15; Line No. 1). We also made figure 4 into figure 4 and figure 5 for better understanding.
[Second review] The term nanocomposite is not appropriate as well. Only the silver nanoparticles in the composite materials are proven to be of nanometric dimensions. The calculated crystallite size does not reflect the actual particle size.
Second Response: TEM image of ZnO and Ag exhibits nanodimensionality. ZnO looks like aggregated particles, which is common with biosynthesized ZnO nanoparticles.
ZnO biological synthesis using Urginea epigea has similar morphology (https://www.sciencedirect.com/science/article/pii/S2405844022035319#fig5) and Larichev (2022) (https://www.mdpi.com/2304-6740/10/12/248) results are also similar to our results. So, we would like to retain the title nanocomposites.
- In Figure of photodegradation you mention mineralization of methylene blue, it is also showcased in Figure 5b. However, you never confirmed it. For such a statement an additional test should be performed, i.e., total organic carbon analysis. Methylene blue may be degraded to smaller molecules not absorbing light in the red region of the spectrum. Also, the C0should be the concentration of MB after the dark period of adsorption-desorption. I did not find the information whether the MB solution was irradiated without the presence of the photocatalyst – this could also impact the results, as MB undergoes photobleaching by itself as well. As a result, the ‘Degradation’ as shown on the y axis in Figure 6d should be changed to ‘Removal’. Additionally, this part could also use some toxicity assessment of the post-degradation solution.
Response (1): We performed only elemental analysis of ZnO and Ag/ZnO, and the results of the weight% and atomic% of carbon content were estimated through SEM-EDX and TEM-EDX analysis. In the presence of a catalyst, the cationic thiazine dye, MB was reduced into less toxic colorless leuco-methylene blue (LMB). Many previous reports explain that MB gets degraded into LMB with a decrease in the peak at 295 nm. We also performed the degradation of MB without catalyst, which showed that the MB concentration was slowly reduced by photoreaction i.e., degradation of 25.8 ± 1.91% after 180 min (Supplementary Figure 3) (Page No. 24, Line No.3-5; 14-18). We are planning to perform a full-fledged phototoxicity analysis with various degradation analyses as our next project.
Supplementary Figure 3. Photolysis of methylene blue dye without any photocatalysts.
[Second review] Thank you for the clarification. Based on the information given I am not sure whether the biosynthesized ZnO is photoactive. As the photodecomposition of MB alone is around 25% and the adsorption is responsible for around 20% decrease in MB concentration, This leaves about 5% that is photodegraded, as the Ct/C0 after 120 min is ~50%. The description of the figure and in the main text also was not changed from photodegradation to removal which is more appropriate. Also, as the LMB was not confirmed only assumed, I would change “was reduced into the less toxic colorless leuco-methylene blue (LMB)” to “probably was transformed into the less toxic colorless leuco-methylene blue (LMB)”.
Second Response: Biosynthesized ZnO is less photoactive because of the presence of biomacromolecules (which is evident from the presence of carbon from the elemental analysis), which hinders photocatalysis. However, Ag/ZnO is comparably more photoactive due to the probable mechanism mentioned in Figure 7b. The figure caption was rewritten as probable mechanisms of photocatalysis. Also, we have changed the sentence mentioned, “Figure 7b depicts the most probable photocatalytic mechanism…” (Figure 7 caption; Page No. 25, Line No. 18). We have also corrected the degradation percentage based on the C0 value as the concentration after adsorption/desorption equilibrium and the corresponding degradation percentages (Page No. 24, Line No. 15, and Figure 9).
- The 3.3.1.1 section is based on literature only. You assume that the mechanism is what you described but additional experiments should be performed to confirm it – for example EPR experiments or photodegradation experiments with ROS quenchers and how it affects the photodegradation efficiency.
Response (1): Thank you. We removed section 3.3.1.1. and merged it with 3.3.1. section. Yes, this is the proposed mechanism of degradation, and it has been already proven by many researchers with similar silver/zinc oxide nanocomposites [65-67]. The importance of this composite is its biological synthesis and established the result of having the potency to involve in photodegradation, antimicrobial, and anticancer results.
[Second review] If this is the case, that the mechanism is well established, I do not see the need to present the whole mechanism in the paper. On the other hand – can you be sure that the use of CFF does not alter the mechanism of the photodegradation and ROS involved?
I do not understand the calcination of the material between the cycles – it is a rather non-standard procedure, as a photocatalyst should be effective without such treatment between cycles. Can you cite any references where such treatment was performed?
Second Response: Here, CFF is not used as such for photodegradation; it is only mediating the synthesis of Ag/ZnO. Washing several times with water and alcohol and calcination at 450 °C removes almost all major biomacromolecules, except which is used to stabilize the synthesized materials. So, there could not be any possibility for the CFF to involve in the photodegradation directly, and no such reports were available. However, previous reports explain that the bacterial filtrate exhibits antioxidant activity (https://bmcmicrobiol.biomedcentral.com/articles/10.1186/s12866-017-1050-2). So, we can neglect the possibility of ROS production by CFF in photodegradation.
In the recycling experiment: We regenerated the Ag/ZnO by drying it at 100 °C for 1 h after washing it in ethanol; there was only an indiscernible decrease of 7.3% in the degradation from the initial degradation capability (supplementary figure 4) (Page No. 25, Line No. 13-15).
- The anticancer part – to consider these materials as anticancer agents, experiments with healthy (non-cancerous) cells should be provided. Also, the big size of the particles (>1000 nm in DLS) clearly show the poor prognosis of potential clinical application – particles of such huge size would not reach the target tissues/cells.
Response (1): DLS results are the qualitative result, they vary with changes in the morphology of the nanoparticles/nanocomposites. TEM is the only possible confirmatory experiment to confirm the size of nanoparticles. TEM shows the formation of Ag NPs with a size of 19.0 ± 11.9 nm attached to the submicronic, irregularly shaped, puffy-like ZnO particles. Nanoparticles with a diameter of 10-100 nm are considered efficient anticancer agents due to their effective delivery and enhanced permeability and retention (EPR) effects. Smaller particles are considered to undergo rapid release from normal vessels and damage healthy cells and tissue before their metabolism by the kidneys (<10 nm in size) [68]. Moreover, anticancer activity can be exhibited by local administration on the targeted site for photodynamic therapy. These composite materials act as a reservoir of releasing metal ions inducing oxidative stress required for the inhibition of the cancer cell's growth and death (Page No. 27, Line No. 10-13)
[Second review] The particle size of 19 nm refers to silver, which is deposited on the surface of ZnO, which are considerably larger as seen in the micrographs and the only values of the particle size are the results from DLS. The DLS results are affected by a series of factors, however they show how big can the particles/agglomerates be regarded in aqueous medium (hydrodynamic diameter). SEM and TEM only show the actual particle size without the effect of the solvent or agglomeration. And particles >1000 nm, regardless of whether these are individual particles or agglomerates, would trigger macrophage immune response and be quickly removed from bloodstream. If topical application is regarded such information should be added to the limitations of the study.
The healthy cells experiments would show that there are no leftover cellular elements present on the surface of the materials that would act in a cytotoxic manner. It should be performed.
Second Response: Thank you. We agree that various factors greatly influence DLS results, and our results with >1000 nm are due to their hydrodynamic size. Depending on the property of the nanocomposite, it can be used in systemic or local administration. In our study, since the agglomeration in water is comparatively higher, it cannot be used in systemic administration for biomedical applications, which also needs targeting of cancer cells. In that case, local administration can be done. Our nanoparticle is probably a potential candidate for local administration rather than for systemic administration for biomedical applications (Page No. 29, Line No. 13-15).
- Please describe in the Figure caption the regions highlighted in Figure 2d, as it is unclear without reading the whole article.
Response: Thank you, we have elaborated on the peaks in Figure 2d (Figure 3b).
[Second review] Thank you for the changes made.
- There seems to be a discrepancy in the description of the scale bars in the Figure 3 caption and in the scale, bars present in the figure itself.
Response: Thank you. Yes, we have corrected the scale bars for Figure 3 (Figure 4).
[Second review] Thank you for the changes made.
- What are the unmarked reflections at ~28 and 35° in the XRPD?
Response: XRD peaks at 28 and 45 are probably from the macromolecules present in the CFF (Page No. 14, Line No. 2).
[Second review] Is it possible that the macromolecules were still present in their crystalline form after 450°C treatment?
Second Response: metals possible from the contents of the biomacromolecules, which were in close interaction with the metal or metal oxide particles.
- It is weird to me that you mention that ‘various physical and chemical methods (…) are expensive and require sophisticated instruments’ and at the same time rely on photodegradation. This seems contradictory. Also, the Fenton process and photodegradation are types of AOPs and here it seems that they are separate methods.
Response: Thank you for mentioning the wrong statement. We have modified the statement for readers' clear understanding (Page No. 3, Line No. 16-22).
[Second review] Thank you for the changes made.
- Two authors have no affiliations: Rakesh Bhaskar and Yong Joo Seok. There are also some of the CReDiT statements missing, most notably investigation.
Response: Thank you for correctly pointing out the mistake. We have added the missing affiliation data for the authors.
[Second review] Thank you for the changes made.

Reviewer 2 Report
I acknowledge that the authors have made certain revisions and improvements in response to the main issues I raised during my last review. However, in my perspective, this manuscript still does not meet the standards I would deem fit for publication.
That said, I do understand that perceptions regarding the readiness of a paper for publication may vary among reviewers and editors. If you, along with other reviewers, believe this manuscript has reached a publishable level, I will not strongly object. My intention is to encourage the refinement of the scientific quality of the paper to the greatest extent possible, rather than rigidly preventing its publication.
Author Response
Comments and Suggestions for Authors
I acknowledge that the authors have made certain revisions and improvements in response to the main issues I raised during my last review. However, in my perspective, this manuscript still does not meet the standards I would deem fit for publication.
That said, I do understand that perceptions regarding the readiness of a paper for publication may vary among reviewers and editors. If you, along with other reviewers, believe this manuscript has reached a publishable level, I will not strongly object. My intention is to encourage the refinement of the scientific quality of the paper to the greatest extent possible, rather than rigidly preventing its publication.
Response: We thank the reviewer for their comments to improve the quality of the paper; we have largely modified the manuscript from the initial submission. In this 2nd revision, we have also rectified many scientific issues and improved the English for readers to understand. All the modifications are highlighted in the manuscript.

Reviewer 4 Report
It's better now.
Please improve figures 3,4,5,6 and 8.
Proofreading is required.
Author Response
Comments and Suggestions for Authors: It's better now.
Please improve figures 3,4,5,6 and 8.
Response: Yes, we have improved the quality of figures 3,4,5,6, and 8 (>800 dpi) and also made proper figure captions for readers’ clear understanding. Figures 2, 5, 7, and 9 are modified.
Comments on the Quality of English Language: Proofreading is required.
Response: Thank you. We have improved the English language with the aid of an English editing service.

Reviewer 6 Report
This version can be accepted for publication
Author Response
Comments and Suggestions for Authors: This version can be accepted for publication
Response: Thank you for your valuable suggestion for improving the manuscript.
